# Microbial Trojan Horses: Virulence Factors as Key Players in Neurodegenerative Diseases

**DOI:** 10.3390/molecules30030687

**Published:** 2025-02-04

**Authors:** Matheus V. C. Grahl, Kelvin Siqueira Hohl, Thiago Smaniotto, Célia R. Carlini

**Affiliations:** 1Graduate Program in Medicine and Health Sciences, Pontifical Catholic University of Rio Grande do Sul, Porto Alegre 90619-900, RS, Brazil; 2Graduate Program in Biochemistry, Institute of Health Basic Sciences, Federal University of Rio Grande do Sul, Porto Alegre 90035-003, RS, Brazil; kelvin.hohl@outlook.com (K.S.H.); tasmaniotto@hotmail.com (T.S.); 3Center of Biotechnology, Department of Biochemistry, Federal University of Rio Grande do Sul, Porto Alegre 91501-970, RS, Brazil; 4Graduate Program of Biosciences, Federal University of Health Sciences of Porto Alegre, Porto Alegre 90050-170, RS, Brazil

**Keywords:** gut–brain axis, outer membrane vesicles, virulence factors, neurodegenerative disease

## Abstract

Changes in population demographics indicate that the elderly population will reach 2.1 billion worldwide by 2050. In parallel, there will be an increase in neurodegenerative diseases such as Alzheimer’s and Parkinson’s. This review explores dysbiosis occurring in these pathologies and how virulence factors contribute to the worsening or development of clinical conditions, and it summarizes existing and potential ways to combat microorganisms related to these diseases. Microbiota imbalances can contribute to the progression of neurodegenerative diseases by increasing intestinal permeability, exchanging information through innervation, and even acting as a Trojan horse affecting immune cells. The microorganisms of the microbiota produce virulence factors to protect themselves from host defenses, many of which contribute to neurodegenerative diseases. These virulence factors are expressed according to the genetic composition of each microorganism, leading to a wide range of factors to be considered. Among the main virulence factors are LPS, urease, curli proteins, amyloidogenic proteins, VacA, and CagA. These factors can also be packed into bacterial outer membrane vesicles, which transport proteins, RNA, and DNA, enabling distal communication that impacts various diseases, including Alzheimer’s and Parkinson’s.

## 1. Introduction

The increase in quality of life in recent decades has extended life expectancy and favored population growth. Estimates from the WHO point to approximately 2.1 billion people over 60 years old by 2050. Because of this increase, the age pyramid of society is undergoing a process of inversion. The proportion of young people has decreased while there has been a significant increase in the proportion of the elderly. With the rise in the elderly population, neurodegenerative diseases (NDs) are becoming more frequent [1].

NDs are disorders caused by the progressive loss of neuronal cells or their function. These diseases can arise from various causes: incorrect protein folding, immunological alterations, oxidative stress, genetic changes, neuronal apoptosis, and interactions of the gut–brain axis [2,3,4]. Unfavorable gut–brain interactions occur when the gut microbiota undergoes a process of imbalance, culminating in the proliferation of potentially harmful microorganisms that can lead to inflammatory reactions and the permeabilization of the intestinal epithelium. Moreover, metabolites produced by the altered microbiota eventually affect the integrity of the intestinal barrier, reach the systemic circulation, and even disrupt the blood–brain barrier (BBB) [5].

The microorganisms composing the microbiota can produce various virulence factors for their self-protection against the host’s defenses. These virulence factors can play an important role in the exacerbation of ND. One example is the production of urease by the bacterium *Helicobacter pylori*. Recent studies revealed that the systemic administration of this protein induces a (neuro)inflammatory response in rodents and modifies the phosphorylation status of Tau, a hallmark of Alzheimer’s disease (AD) [6]. Other bacterial virulence factors also could potentially contribute to ND. Bacterial lipopolysaccharides (LPSs) and proteins of the outer membrane of Gram-negative bacteria appear in elevated levels in the plasma of AD patients as compared to a control group [7]. LPSs have been implicated in the formation of extracellular amyloids [8,9].

Bacterial extracellular vesicles can carry virulence factors distal to the primary infection site, establishing host–pathogen cellular communication. Outer membrane vesicles (OMVs) contain various virulence factors, such as outer membrane proteins, periplasmic proteins, RNA, and DNA, which are enclosed in OMVs during the formation process [10]. Bacterial OMVs have been associated with various pathologies, such as cancer, pulmonary diseases, and NDs [11,12,13].

Considering the need to understand the mechanisms by which NDs are triggered, this review aims to describe the current landscape of the potential contributions of microbial virulence factors to the pathogenesis or progression of NDs. We seek to establish an understanding of how these molecules can aid in the development of NDs and if they can be considered molecular targets for the treatment of these conditions. Furthermore, we aim to summarize the current available treatment options for ND-related microorganisms.

## 2. Neurodegenerative Conditions Focused in This Review

### 2.1. Alzheimer’s Disease

Alzheimer’s disease (AD) accounts for 60% of dementia cases in the elderly worldwide [14,15]. It is estimated that in 2022, around 55 million people will be living with dementia [1]. In 2019, the global cost of dementia treatment was estimated to be around USD 1.3 trillion per year [1].

AD is characterized by dementia, memory deficits, behavioral changes, and the inability to perform daily activities [16,17]. Physiologically, AD is marked by changes in glial cells and the presence of senile plaques and neurofibrillary tangles in the brain tissue. The plaques are made up of deposits of β-amyloid (Aβ) peptides, and the tangles are formed by the deposition of hyperphosphorylated Tau proteins (Tau-p) (Figure 1) [18].

The amyloid precursor protein (APP) is a transmembrane protein found in neuronal synapses that is cleaved by secretases, producing different proteolytic fragments. In the non-amyloidogenic pathway, APP is cleaved by α-secretase (composed of presenilins 1 (PSEN1) and presenilins 2 (PSEN2)), generating two proteolytic fragments, extracellular APPsα and αCTF (C-terminal fragment, 83 amino acids residues), which are subsequently cleaved by λ-secretase, producing p3 and the amyloid precursor protein intracellular domain (AICD). Mutations in the PSEN1 and PSEN2 genes favor the excessive production of Aβ42, accelerating the formation of amyloid plaques, which is directly related to neurotoxicity and disease progression [19].

The production of APPsα and αCTF prevents the formation of Aβ [20]. In the amyloidogenic pathway, APP is cleaved by β-secretase, generating APPsβ and βCTF (99 amino acid residues). The intramembrane cleavage of βCTF by λ-secretase yields Aβ peptides (38–42 amino acids) and AICD. Aβ peptides are released to the extracellular space as monomers [21]. These peptides are generated at high levels by synaptic activity and are highly prone to aggregation, especially Aβ42. When oligomerized, the Aβ peptide interacts with various receptors, such as NMDA, metabotropic 5, and acetylcholine α7, causing pathological changes in dendritic spines and synaptic efficiency [22].

AD also involves pathophysiological processes mediated by an apolipoprotein (ApoE). ApoE interacts with Aβ monomers, directing them to low-density lipoprotein receptors (LDLRs) and LDL receptor-related protein 1 (LRP1), targeting the monomers to the perivascular space of arterioles and venules, where they are subsequently removed via the glymphatic pathway, lowering the amount of available Aβ [23]. However, when certain ApoE isoforms are present (ApoE2, ApoE3, and ApoE4), the transport of Aβ is inhibited, decreasing the clearance of the peptide [24]. ApoE4 can also play a crucial role in Aβ clearance through endosomal alterations, blocking the release of exosomes and permeabilizing the lysosome, leading to the release of cathepsins that activate cell death cascades [25,26,27,28]. Notably, once released, cathepsin B facilitates the assembly of the NLRP3 inflammasome and activates caspase 1, leading to the synthesis of pro-inflammatory IL-1β [29].

The Tau protein is essential for the functional organization of neurons through the stabilization of cytoplasmic microtubules. Six isoforms of Tau protein are resultant from exon splicing [30]. Tau is physiologically phosphorylated by diverse kinases. On the other hand, the hyperphosphorylation of Tau disrupts its interaction with microtubules, rendering them unstable. Tau-p undergoes conformational changes, loses its biological functions, becomes resistant to degradation, and is insoluble and prone to aggregation [31]. Tau-p is then released to the extracellular space by synaptic activity and internalized (endocytose) by postsynaptic neurons and glial cells [32,33]. Tau-p is subsequently deposited in the cell bodies and dendrites of neurons, forming fibrillary tangles [34,35]. These protein changes, along with the other factors mentioned earlier, progressively lead to the death of nerve cells. The most affected areas in the brain are the hippocampus and the cerebral cortex, resulting in the pronounced atrophy of these regions [36].

The regulation of Tau is mediated by phosphorylation and dephosphorylation. Approximately 80 phosphorylation sites have been described for Tau. Glycogen synthase kinase 3-β (GSK-3β) is the main enzyme responsible for maintaining the stability of this protein. As described in the literature, there is a pathogen-mediated relationship between the activation of GSK-3β, the hyperphosphorylation of the Tau protein, and the progression of AD. As reported by Wang et al. (2015) [37], the gastric pathogen *H. pylori* can induce the hyperphosphorylation of Tau through the activation of GSK-3β.

Other factors are also associated with AD, such as hyperactivity of the calcium flow around the plaques (which can be normalized by Aβ inhibition). The presence of Aβ plaques and the increase in intracellular calcium can trigger reactive astrogliosis. The elevation of intracellular calcium activates NADPH oxidase and inducible nitric oxide synthase (iNOS), leading to the production of reactive oxygen species (ROS) and nitric oxide (NO) in astrocytes, which may culminate in the death of these cells [38].

Moreover, reactive astrocytes can inhibit phagocytosis by microglia and contribute to glutamatergic dysregulation. The increase in glutamate exacerbates calcium influx into cells, creating a positive feedback loop that further intensifies glutamate release [38].

Microglia surrounding Aβ aggregates proliferate but are unable to phagocytize them, resulting in a chronic inflammatory state mediated by interleukins such as IL-1β, IL-6, and TNF-α. These interleukins overactivate adjacent glial cells, perpetuating a pro-inflammatory activation state [39,40,41]. In the same way as astrocytes can induce microglial activation, a reciprocal effect can also be observed. Reactive microglia are capable of stimulating astrocytes to become immunologically active, promoting the release of inflammatory mediators such as TNF-α, TGF-β, IL-1β, and IL-6 [38].

Abnormal clearance through the endosomal–lysosomal complex and the βCTF-induced hyperactivation of Rab5 (a GTPase) cause an increase in endosomes and consequently endocytosis, resulting in changes in synaptic plasticity; contribute to the hyperphosphorylation of Tau; and cause neurodegeneration [42]. Conversely, autophagy induced by lysosomes is also affected due to the accumulation of Aβ and βCTF [43].

During the 1970s, a cholinergic death hypothesis was proposed for AD. Acetylcholine is involved in processes of sensory information, learning, memory, and attention. It is synthesized in cholinergic neurons from choline and acetyl-CoA by the enzyme choline acetyltransferase. Acetylcholine is delivered at the synaptic vesicles by a specific protein, the vesicular acetylcholine transporter. It was hypothesized that the damage caused by Aβ fibrils affects the enzymatic activity of acetylcholinesterase and reduces choline uptake and acetylcholine release, resulting in the death of cholinergic neurons accompanied by the loss of cognitive and memory functions [44]. On the other hand, AD also appears to be related to a decrease in nicotinic and muscarinic receptors (M2) and a reduction in glutamate concentration and D-aspartate uptake [45].

### 2.2. Parkinson’s Disease

Parkinson’s disease (PD) is the second leading ND. As a result of neuronal loss, this pathology leads to motor symptoms such as tremors, rigidity, and postural abnormalities. Additionally, non-motor symptoms are observed, including anxiety, depression, cardiovascular complications, urological issues, olfactory disturbances, and gastrointestinal problems [46]. Currently, it is estimated that the etiology of this disease is multifactorial.

PD is biochemically characterized by the presence of Lewy bodies, which are inclusions that are approximately 5 to 30 µm in diameter, and they are found in the cytoplasm of neuronal cell bodies in brain structures such as the substantia nigra (dopaminergic), raphe nuclei (serotonergic), and locus coeruleus (noradrenergic) (Figure 2). Lewy bodies contain the presynaptic protein α-synuclein (~140 amino acids) in a fibrillar state (5–10 nm in length) that is aggregated or phosphorylated (Ser129) [47]. Moreover, other components may be present in Lewy bodies, such as Tau, oxidized/nitrated proteins, parkin, ubiquitin, heat shock proteins, neurofilaments, microtubule-associated proteins (MAPs), tubulin, and proteasomal and lysosomal elements [48].

Functional α-synuclein acts as a chaperone, assisting in the assembly of the SNARE complex, and it is involved in the release of dopamine [49]. This protein consists of a conserved N-terminal (positions 1 to 60) region with an amphipathic α-helix structure that allows interactions with membranes and a polar C-terminal region (positions 96 to 140) where post-translational modifications and protein–protein interactions occur. The central region of α-synuclein (positions 61 to 95) carries its amyloidogenic property [50].

In general, phosphorylated α-synuclein is found in the spinal cord and cervical and thoracic sympathetic ganglia [51]. Different conformations of α-synuclein exist throughout the body. Accumulated α-synuclein can be found in various peripheral sites such as the cardiovascular and the gastrointestinal (stomach and intestines) systems [52,53]. β sheet-enriched α-synuclein is more prone to aggregation, and various factors may be associated with this process, such as phosphorylation at Ser129, C-terminal truncation, and ubiquitination [54]. These alterations lead to different conformational states of the protein, ranging from fibrils to oligomers, with the latter being the more harmful [55].

Some genes related to the ubiquitination system appear to be associated with PD. *PARK2* and *PARK5* are genes related to E3 ubiquitin ligase and C-terminal ubiquitin hydrolase, respectively; alterations in these genes lead to the accumulation of α-synuclein and subsequent neuronal death [56,57]. On the other hand, the autophagy system also presents genetic alterations linked to PD. This includes the *PARK9* gene encoding the lysosomal protein ATP13A2, the *PARK2* gene encoding parkin, and the *PARK6* gene, which is translated into the PINK1 protein. When PINK1 degradation is reduced, this protein accumulates on the outer mitochondrial membrane, resulting in the recruitment of parkin, which activates E3 ubiquitin ligase, signaling autophagy [56,58,59].

Post-mortem analyses of brains from PD patients have shown an increase in phosphorylated Tau at Ser262, Ser396/404, and Aβ accumulation in the striatum [60,61,62]. Furthermore, in vivo and in vitro analyses reinforced the possible interaction between α-synuclein and phosphorylated Tau [63].

Dopaminergic neurons are the primary cells affected during PD. Dopamine is produced from the amino acid tyrosine, which is internalized into the cells via transporters, and then converted to dihydroxyphenylalanine (L-DOPA) by tyrosine hydroxylase. Subsequently, L-DOPA is decarboxylated to produce dopamine, which is internalized into synaptic vesicles via vesicular monoamine transporter 2, preventing its autoxidation [64].

During the metabolism of dopamine, it can undergo autoxidation in its catechol rings, producing O_2_•− and H_2_O_2_. Once these compounds are formed, they may interact with other components such as •NO and iron (PD patients have elevated levels of iron), respectively [65]. The products of these reactions generate free radicals (ONOO− and •OH, respectively), causing damage at various cellular levels through cytotoxic reactions, incorrect protein folding, lipid peroxidation, activation of glial cells, mitochondrial dysfunction, α-synuclein aggregation, and reduced proteasomal activity [65]. On the other hand, neuromelanin, a metabolite of dopamine and norepinephrine, can chelate iron and neutralize free radicals, generating a protective effect. A similar effect occurs with glutathione, which reduces free radical levels [50].

PD presents neuroinflammatory aspects, such as increased innate immune responses, with the elevated expression of TLR2, TLR5, TLR4, and CD14; activation of natural killer cells; and increased levels of IL-1β, IL-6, TNF-α, IL-2, IL-4, interferon-λ, and complement factors C3 and C9 [66,67]. Microglial activation appears to play an important role in this disease. Microglia, the main MHC II antigen-presenting cells in the brain, contribute to the pathology through the release of pro-inflammatory factors and the increased production of NO, leading to α-synuclein aggregation [68].

Once α-synuclein is present in peripheral regions, the uptake of this protein by dendritic cells can lead to the peripheral activation of MHC II cells, stimulating the production of CD4^+^ T cells, while activation of MHC I cells leads to activation of CD8^+^ T cells. Both cell types have been found in the post-mortem studies of PD patients, accompanied by a reduced number of dopaminergic neurons and increased levels of pro-inflammatory cytokines in brain tissues [69]. Stimulated CD4^+^ cells can differentiate into Treg, Th2, Th1, and Th17, while CD8^+^ cells differentiate into cytotoxic T lymphocytes. These cells can migrate through the bloodstream and reach the brain due to the leakage of the BBB in the inflamed endothelium [69].

In the brain, immune cells are stimulated again when they are exposed to α-synuclein epitopes presented by MHC II on microglia and MHC I on neurons. Hence, astrocytes and microglia suffer damage from Th1/Th17 cells associated with the production of interferon-λ and IL-17. The activation of astrocytes leads to the production of IL-6 and activated microglia release IL-1α, TNFα, C1q, and NO. These molecules eventually lead to the death of dopaminergic neurons [69].

The adaptive immune response also plays an important role in the pathology, through the increase in T cells and antibodies against neural antigens [66,67].

Previous studies have indicated a possible role of neuroinflammation and the gut–brain interactions in PD. A high prevalence of *H. pylori* or *Proteus mirabilis* infection in PD patients, with the detection of these pathogens in stool samples, may indicate that intestinal dysbiosis triggers or contributes to this pathology [70,71]. Choi et al. (2018) demonstrated that after administering *P. mirabilis* by the oral route to mice for 5 consecutive days, the animals showed motor alterations characteristic of PD, accompanied by the loss of dopaminergic neurons, the presence of neuroinflammation markers (Iba1, TNF-α, and IL-1β), and increased deposits of α-synuclein [46]. Complete vagotomy in PD patients resulted in a reduction in disease progression compared to partial vagotomy or no vagotomy, reinforcing the involvement of the vagus nerve and gut–brain interactions in PD [72,73].

## 3. Microorganisms Involved in Neurodegenerative Diseases

### 3.1. Gut–Brain Axis

#### 3.1.1. *Prevotellaceae*

*Prevotellaceae* are a group of anaerobic bacteria commonly found in the human gut microbiome. Changes in the abundance of *Prevotellaceae* have been observed in both AD and PD. The increased abundance of *Prevotellaceae* in AD has been reported, suggesting a potential role of bacteria in this family in the disease’s pathogenesis. Similarly, in PD, a higher abundance of *Prevotellaceae* was reported, indicating that a shift in gut microbiota composition favoring these bacteria may contribute to the development and progression of the disease [74,75,76].

The overgrowth or dysbiosis of *Prevotellaceae* has been linked to the development of inflammatory conditions and the potential exacerbation of ND [77]. However, the mechanisms by which *Prevotellaceae* may influence the onset and progression of AD and PD are not fully understood, but several hypotheses have been proposed. *Prevotellaceae* may contribute to the aggregation of amyloid proteins (β-amyloid and α-synuclein), which are hallmark features of AD and PD, respectively. Additionally, metabolites and inflammatory mediators derived from *Prevotellaceae* may impact the gut–brain axis, leading to neuroinflammation and neurodegeneration through the release of virulence factors [75,76,77,78].

Supporting these findings, the increased abundance of *Prevotellaceae* may have detrimental consequences in ND. These mechanisms include alterations of the integrity of the intestinal and blood–brain barriers, resulting in greater permeability and infiltration of inflammatory molecules into the brain. This, in turn, may promote neuroinflammation and contribute to the neurodegeneration observed in these diseases [75,76,77,79].

Among the bacteria in this genus, certain species stand out: *Prevotella corpi*, which participates in the degradation of complex polysaccharides and the production of short-chain fatty acids (SCFAs) with anti-inflammatory properties [80,81,82]; *Prevotella melaninogenica*, associated with protein and carbohydrate degradation, also having the capacity to produce SCFAs [81,82]; *Prevotella brevis*, playing an important role in polysaccharide fermentation and SCFA production [83,84]; *Prevotella intermedia*, involved in the breakdown of proteins and complex carbohydrates [84,85]; and *Prevotella nigrescens*, which also has a role in the fermentation of complex carbohydrates [85].

Thus, disturbances in the increase or decrease in these species lead to gut dysbiosis, which in turn can result in severe and often irreversible damage. This favors the development and progression of systemic and central damage.

#### 3.1.2. *Lactobacillaceae*

The family *Lactobacillaceae* is composed of lactic acid-producing bacteria that have increasingly been associated with ND, such as AD and PD. These bacteria are known for their diverse metabolic capabilities and a wide range of applications in the food and feed industries.

Certain species of the *Lactobacillaceae* family have been observed at elevated levels in individuals with AD and PD [86,87,88]. Changes in the gut microbiome, including shifts in the abundance of *Lactobacillaceae*, have been linked to the aggregation of α-synuclein and the formation of β-amyloid plaques, hallmark features of AD and PD [76,79].

The potential mechanisms by which *Lactobacillaceae* may contribute to the development and progression of these ND include the ability of these bacteria to adjust the pH (through the enzymatic activity of urease acting as the virulence factor) of the intestinal environment, creating favorable conditions for the growth of pathogenic bacteria that can damage the intestinal barrier [89,90]. Like other bacterial family, *Lactobacillaceae* can contribute to increased intestinal permeability, allowing the release of virulence factors into the systemic circulation and enabling them to migrate to the central nervous system (CNS) [90,91].

Species of *Lactobacillaceae* participate in various functions, such as producing short-chain fatty acids (SCFAs) and neurotransmitters, which can have both beneficial and harmful effects on the host. For example, some strains of *Lactobacillus* have demonstrated anti-inflammatory properties and may help maintain intestinal barrier integrity [75,76]. However, other *Lactobacillus* species may promote inflammatory pathways and compromise the intestinal epithelial barrier [75,76,92].

Among the species in this family, *Lactobacillus acidophilus* stands out for its probiotic properties, aiding in lactose digestion and lactic acid production and maintaining gut flora through pH regulation [93,94]. *Lactobacillus rhamnosus* is used in the treatment of diarrhea, urinary tract infections, and intestinal issues such as irritable bowel syndrome due to its ability to adhere to the intestinal epithelium, promote barrier integrity, and produce antimicrobial compounds that inhibit pathogens [95,96]. *Lactobacillus plantarum* has antioxidant properties, helping reduce inflammation by balancing the microbiota. This activity is related to its ability to ferment carbohydrates and fibers, contributing to the production of SCFAs like acetate, which has anti-inflammatory properties [97]. *Lactobacillus reuteri* is known for producing antimicrobial compounds, such as reuterin, which help control the growth of pathogens [98,99].

Although the *Lactobacillaceae* family includes beneficial bacterial species, some can also be harmful. For instance, *Lactobacillus gasseri*, when its colony increases, can promote intestinal inflammation [100]. *Lactobacillus johnsonii* is known to contribute to opportunistic infections, potentially translocating from the intestine to the bloodstream in immunocompromised individuals [101]. *Lactobacillus delbrueckii* has been associated with lactic acidosis, a condition in which the local environment becomes excessively acidic, compromising the function of tissues and organs [102,103].

#### 3.1.3. *Bacteroidaceae*

*Bacteroidaceae* are a family of Gram-negative anaerobic bacteria commonly found in the human gastrointestinal tract.

Some species within the *Bacteroides* genus are capable of promoting benefits to the organism. For example, *Bacteroides fragilis*, *Bacteroides vulgatus*, and *Bacteroides dorei* help in maintaining intestinal barrier integrity, digesting complex polysaccharides, and producing SCFAs like butyrate, which has anti-inflammatory properties [81,98]. *Bacteroides thetaiotaomicron* is important in the breakdown of complex carbohydrates, such as dietary fibers, and in the production of SCFAs, acting as a facilitator of gut health [104,105].

#### 3.1.4. *Enterobacteriaceae*

Bacteria from the *Enterobacteriaceae* family can have a detrimental impact by contributing to systemic inflammation and gut microbiome dysbiosis, which are strongly associated with neuroinflammatory processes and neurodegenerative diseases [106]. *Escherichia coli* can be present as part of the gut microbiome, and commonly, it is associated with urinary tract infections; more recently, it was found as a possible cause of amyloidogenesis [107]. The release of a variety of virulence factors allows for increased intestinal permeability, followed by the release of proteins into the bloodstream. Once in circulation, these proteins can migrate to the nervous system due to a weakened blood–brain barrier and contribute to the progression of a neuroinflammatory state. The entry of these bacterial components into the brain can activate Toll-like receptor 4, contributing to chronic neuroinflammation and potentially accelerating the progression of conditions such as Alzheimer’s and Parkinson’s [92].

Urinary tract pathologies are frequent among elderly patients with cognitive impairment. This is the case observed for *Proteus mirabilis*, *Klebsiella pneumoniae*, and *Enterobacter cloacae*, organisms associated with urinary tract infections that also contribute to neurodegenerative diseases (NDs), such as Parkinson’s disease [107].

*P. mirabilis*, when administered to mice, was able to induce motor deficits associated with PD. Choi et al. (2018) demonstrated that gut dysbiosis could induce damage to dopaminergic neurons in the striatum, resulting in motor impairment. Furthermore, the authors showed an increase in the presence of aggregated α-synuclein in the brain and in the colon. Additionally, it was described that this species may contribute to increased central inflammation, corroborating the observed damage [46].

*K. pneumoniae* is a pathogenic bacterium associated with severe infections such as pneumonia, urinary tract infections, and sepsis. This bacterium can colonize the gastrointestinal tract and induce inflammation. The gut dysbiosis and systemic inflammation induced by *K. pneumoniae* may worsen neuroinflammation, particularly in Parkinson’s disease [108,109]. The inflammatory response generated by this pathogen compromises the intestinal barrier, allowing toxins to reach the brain and promoting neurodegeneration [110].

*E. cloacae* is another opportunistic pathogenic species associated with inflammation. Studies have reported higher levels of *E. cloacae* in patients with neurodegenerative diseases, such as Parkinson’s disease [111]. Certain strains of *E. cloacae* can trigger an inflammatory immune response and produce LPS, activating the TLR-4/NF-κB inflammatory pathway in the brain and contributing to neuronal death and disease progression [112].

Other members of the *Enterobacteriaceae* family, such as *Serratia marcescens*, *Citrobacter freundii*, *Shigella* spp., and *Salmonella* spp., are associated with neurodegenerative diseases. Studies indicate that the exotoxins produced by these organisms can damage cell junctions by cleaving E-cadherin, leading to increased permeability and significantly contributing to the development and progression of neurodegenerative diseases by destabilizing the gut–brain axis and inducing neuroinflammation [46,75,92,107,113].

#### 3.1.5. *Helicobacteraceae*

*Helicobacter pylori* is a Gram-negative bacterium commonly associated with infections of the gastric tract (gastritis, peptic ulcer, and gastric cancer). It is estimated that the pathogen is present in 43.1% of the global population which is mostly assymptomatic [114]. However, the pathogen may also be correlated with other diseases such as peripheral neuropathies, glaucoma, and neurodegenerative disorders. Recent studies link the presence of the pathogen to AD. The detection of anti-*H. pylori* immunoglobulins and homocysteine in the serum of AD patients occurred in 88% of patients with the disease compared to 46.7% in the control group [115].

Other studies indicate that 100% of AD patients (n = 27) had antibodies against *H. pylori* in both serum and cerebrospinal fluid. Longitudinal studies (20 years) showed that the presence of antibodies to *H. pylori* indicates a 1.46 times higher risk factor for disease progression [116]. Furthermore, studies suggest that treatment of the pathogen may reduce the rate of cognitive decline [117].

*H. pylori* can supposedly reach the brain through the oral–nasal–olfactory route or retrogradely via the vagus nerve. Another infection mechanism involves the use of “Trojan horses,” in which activated monocytes (infected by *H. pylori* due to defective autophagy) can access the CNS through disruptions in the BBB [118].

The metabolites of *H. pylori* can also compromise the CNS, as is the case of outer membrane vesicles (OMVs), which have the ability to cross biological barriers. The bacterium, through its virulence factors like the VacA cytotoxin, exerts chemotactic effects and promotes an increase in the pro-inflammatory response, including the production of TNF-α, IL-8, and ROS. These factors can compromise BBB integrity and intensify neuroinflammation. Additionally, the pathogen also induces the expression of genes related to AD, such as *ApoE2*, *ApoE4*, *APP*, and *TLR-4* [118].

In vitro analyses indicate that the pathogen can induce an increase in presenilin 2, Aβ42, Tau-p, the activation of GSK-3β, and other inflammatory factors such as IL-12, or it can even induce oxidative stress in cell lines [116]. Similar results are observed in vivo, where the intraperitoneal administration of a filtrate of *H. pylori* culture caused memory deficits, increased presenilin 2, Aβ42, and chronic inflammation in rats [37].

*H. pylori*-seropositive AD patients (IgG) have higher levels of Tau and Tau-p (phosphorylated at site 181), and they perform worse in memory tests, psychomotor speed, and orientation, with a reduction in Mini-Mental State Examination (MMSE) scores. The effects caused by the pathogen may occur due to disruptions in nutrient bioavailability from endothelial damage (atrophic gastritis) or through apoptosis triggered by immune factors [117].

The chronic atrophy of the gastric epithelium caused by *H. pylori*-induced gastritis may lead to a decrease in the absorption of vitamin B12 and folic acid, generating secondary hyperhomocysteinemia, which can cause endothelial damage and contribute to AD [115]. Both vitamins play a role in the synthesis of the myelin sheath, and decreased levels may cause memory loss, reversible dementia, and other cognitive impairments.

#### 3.1.6. *Spirochaetaceae*

Spirochetes are helical, Gram-negative bacteria that can have a tropism for the trigeminal nerve/ganglion, leading to acute or latent infections. Contamination by this group of bacteria can occur through hematogenous spread, ascension along the nerve fibers of the olfactory tract, and via the lymphatic system. Patients with AD may present spirochetes in the cerebrospinal fluid, blood, or brain up to 8 times more frequently than control patients. Amyloidogenic proteins from this pathogen can promote the deposition of Aβ plaques [116].

The main microorganisms in this group associated with AD and PD are *Borrelia burgdorferi* and *Treponema pallidum*. *B. burgdorferi* is the causative agent of Lyme disease (borreliosis), while *T. pallidum* is the causative agent of syphilis [107,116].

Patients with AD have been found to present *B. burgdorferi* antigens. Additionally, *B. burgdorferi* can cause cortical atrophy and microgliosis. In vitro analyses showed that after exposure of glial cells and neurons to the pathogen, Tau-p levels increased, leading to microtubule dysfunction, the formation of neurofibrillary tangles, and increased βAPP and Aβ plaques [115]. Meanwhile, ex vivo studies suggest that the pathogen can induce the production of cytokines such as IL-1 in glial cells, IL-6, IL-8, cyclooxygenase-2 (COX-2), and the B lymphocyte chemoattractant chemokine (CXCL13), leading to excessive apoptosis of glia and neurons [116].

Studies suggest that oral infection with *Treponema* was present in 77% of AD patients and in 22% of matched controls without the disease [115]. Syphilis can lead to neurological symptoms such as progressive dementia and memory loss caused by inflammation induced by lipoproteins, amyloidosis, and cortical atrophy [116].

### 3.2. Oral Cavity Bacteria

The oral cavity is the second most populated region by microorganisms after the intestine. Studies point to a correlation between periodontitis and AD due to oral infection, where pathogens primarily start with Gram-positive bacteria, and as inflammation progresses, there is an increase of up to 85% in Gram-negative bacteria [107].

Patients with AD exhibit specific antibodies against periodontal bacteria. Oral infections occur mainly due to *Porphyromonas gingivalis*, *P. intermedia*, *Tannerella forsythia*, *Fusobacterium nucleatum*, *Aggregatibacter actinomycetemcmitans*, *Eikenella corrodens*, and *Treponema denticola*, which are capable of inducing systemic inflammatory responses via TNF-α, IL-1, IL-1β, and IL-6 and a decrease in IL-10, leading to an increase in the C-reactive protein and microglial cells. Inflammation can spread to the CNS through cell-to-cell communication between circulating macrophages and microglia, making them reactive. Another possibility involves the passage of inflammatory factors from the bloodstream across the blood–brain barrier, resulting in neuroinflammation. Additionally, inflammation can also occur via a retrograde pathway through peripheral nerves (via the trigeminal nerve) [116].

Other studies suggest that *Propionibacterium acnes* can affect the CNS through hematogenous dissemination. In AD patients, the presence of *P. acnes* and Treponemas (*T. amylovorum*, *T. denticola*, *T. maltophilum*, *T. medium*, *T. pectinovorum*, *T. socranskii*, and *T. vincentii*) was identified in the frontal cortex [116,119].

Post-mortem studies found LPS from *P. gingivalis* in the brains of AD patients, indicating that periodontal bacteria have the ability to access the brain during life [119]. Studies with APOE mice infected at the gingival level with *P. gingivalis* demonstrated the presence of the pathogen directly in the brain of the animals, in agreement with previously described post-mortem data. During periodontitis, the presence of pathogens in the oral cavity leads to an increase in Aβ (which reflects in increased protein in plasma) and Tau-p, which, combined with APOE allele depletion and increased inflammation, allows molecules to migrate through a compromised blood–brain barrier to the brain, where they may contribute to cognitive impairment and the exacerbation of AD [119].

### 3.3. Pulmonary Microorganism

#### *Chlamydiaceae* 

*Chlamydia pneumoniae* is a Gram-negative, intracellular pathogen. Typically, contamination begins through the nasal or respiratory mucosa and progresses to an acute respiratory infection. However, *C. pneumoniae* infection is not exclusively limited to the respiratory system. Studies suggest that the pathogen may be associated with meningoencephalitis, atherosclerosis, and coronary artery disease [115]. Furthermore, the pathogen can infect and reside inside immune cells and use them to travel to other locations, such as the CNS [120]. Given that the pathogen has an intracellular life cycle, its presence has been detected in various cell types (microglia, astrocytes, macrophages, and monocytes) [121].

Studies have found the pathogen in brain samples from the hippocampus, parietal–temporal lobe, and frontal lobe—areas closely associated with regions rich in neuritic plaques and NFTs in patients with AD (with the presence of *C. pneumoniae* corresponding to about 74–89% of AD-positive patients and the absence of the pathogen occurring in 89–95% of AD-negative patients) [115].

Animal studies indicated that high concentrations of *C. pneumoniae* (4.5 × 10^3^–1 × 10⁸ inclusion-forming units (IFUs)) and low concentrations of *Chlamydia muridarum* are required to induce, via the olfactory route, reactive astrocyte activation, suggesting that the inflammatory response is a process associated with the pathogen infection and may lead to an increase in Aβ plaques [120]. Inoculation of the pathogen directly into the mouse brain demonstrated that the pathogen could cause Aβ plaques and neuroinflammation [115]. *C. muridarum* can generate a Th1 adaptive immune response that is necessary to combat the infection, but it also aggravates tissue damage in the host [122]

However, the results are controversial in the literature. Studies conducted with BALB/c mice demonstrated that after the inhalation of *C. pneumoniae*, the presence of the pathogen was detected only in the olfactory bulb, cerebellum, and hippocampus just one week after administration [121]. However, the authors did not observe any difference in the presence of β-amyloid plaques between treated and control animals over a three-month follow-up period [121].

The difference in in vivo findings may be caused by several factors, including the use of different bacterial strains, strain origin (patient strain vs. ATCC strain), culture medium used, varying concentrations, and even animal strain, all of which complicate comparisons between studies.

## 4. Immune Response in Neurodegenerative Disease Associated with Microbiota Dysfunction

NDs, such as AD and PD, have been increasingly linked to chronic inflammatory processes and dysfunctions in the gut microbiota. The activation of pattern recognition receptors (PRRs), particularly Toll-like receptors (TLRs) and NOD-like receptors (NLRs), plays a central role in mediating the inflammatory response, contributing to the pathogenesis of these conditions [123,124]. The activation of TLR-4, followed by NLRP3 inflammasome activation and subsequent interferon-gamma (IFN-γ) production, suggests that the underlying mechanisms and their implications in the development of ND are intrinsically connected.

Although the mechanisms leading to gut dysbiosis are not fully elucidated, a critical factor involved is the activation of TLR-4 in response to LPS derived from Gram-negative bacteria. LPS can translocate from the gut into the bloodstream during dysbiosis, activating TLR-4 [125]. The interaction of LPS with TLR-4 leads to receptor dimerization and the recruitment of adaptor proteins, such as MyD88 and TRIF. The activation of the MyD88 pathway culminates in the activation of kinases involved in inflammatory signaling, such as the NF-κB transcription factor, which promotes the production of pro-inflammatory cytokines, including TNF-α, IL-6, and IL-1β [126].

Studies have demonstrated that chronic TLR-4 activation is strongly associated with neuroinflammation in ND. For instance, in AD models, TLR-4 activation exacerbates β-amyloid plaque deposition and activates microglia, contributing to neurodegeneration [127]. Thus, the dysregulation of TLR-4-mediated signaling could be a crucial factor in the progression of these neurodegenerative conditions.

Following TLR-4 activation, the inflammatory response can lead to the activation of the NLRP3 inflammasome. This inflammasome is activated by multiple stimuli, including pro-inflammatory cytokines and cellular stress signals. NLRP3 oligomerization results in the recruitment of the ASC adaptor protein and caspase-1, which cleaves the precursors of cytokines IL-1β and IL-18 into their active forms [127]. NLRP3 inflammasome activation has been implicated in several ND, including AD, where its activation is associated with exacerbated neuroinflammation and neuronal death. The release of IL-1β and IL-18 amplifies central nervous system inflammation, promoting microglia activation and the establishment of a neurotoxic environment [124].

IFN-γ, a cytokine primarily produced by CD4^+^ T cells and NK cells, plays a critical role in the adaptive immune response. Its production is induced by the activation of TLR-4 and NLRP3 receptors, playing an essential role in modulating the immune response. In the context of ND, IFN-γ can have a dual effect, promoting both immune defense and neuroinflammation [128]. The action of IFN-γ on macrophages and microglia results in the amplification of the inflammatory response, leading to the production of more inflammatory cytokines and cell death. Excessive IFN-γ production has been observed in AD and PD models, correlating with the severity of inflammation and disease progression [123].

Therefore, the modulation of the IFN-γ signaling pathway and the regulation of TLR-4 and NLRP3 activation represent promising therapeutic targets to reduce neuroinflammation associated with ND.

## 5. Microbial Virulence Components/Factors

### 5.1. Outer Membrane Vesicles

Outer membrane vesicles (OMVs) are derived naturally through budding from the outer cell membrane of Gram-negative bacteria. The characteristics of OMVs vary with growth conditions and bacterial strains. However, the average size is approximately 20–500 nm [129]. OMVs can be generated through different budding mechanisms: (i) loss of cross-links (covalent and non-covalent) between the outer membrane and the peptidoglycan layer; (ii) accumulation of misfolded proteins and cellular fragments in the periplasmic space, leading to increased turgor pressure; (iii) imbalance of divalent cations (e.g., calcium), which cross-link negatively charged molecules such as LPS (this imbalance allows LPS molecules to cause electrostatic repulsion, generating membrane curvatures that facilitate OMV formation); (iv) accumulation of phospholipids in the outer leaflet of the outer membrane due to a deficiency in the ABC transport complex VacJ/Yrb; (v) deacylation of the lipid A component of LPSs [130,131].

In addition to budding, microvesicles can also be formed through explosive lysis. Under cellular stress, often induced by phages, there is an increase in endolysin production, leading to the degradation of bacterial peptidoglycans. This degradation causes the cell to round up and eventually explode. The resulting fragments can reorganize and self-assemble into outer–inner membrane vesicles (OIMVs) or explosive outer membrane vesicles (EOMVs) [130,131].

Regardless of the release mechanism, microvesicles exhibit immunomodulatory properties in the host due to the virulence factors they carry. These characteristics grant OMVs adjuvant properties, making them potential candidates for vaccine-related applications. Furthermore, OMVs can fuse with bacterial membranes, endowing them with important functional properties such as horizontal gene transfer, cellular metabolite export, phage infection, and cell-to-cell communication [130,131].

Their protein/lipid composition is derived from the outer membrane of bacteria and the bacterial periplasmic space/cytoplasm, varying according to the OMV’s release process. They may contain LPS, enzymes, cytotoxins, and nucleic acids depending on the specific bacterium considered (Figure 3). In the case of *H. pylori*’s OMVs, the presence of LPS, urease, the cytotoxins CagA and VacA, and other compounds such as nucleic acids has been reported [132,133]. OMVs exhibit significant heterogeneity in their composition, which can be determined by the genotypes of strains, different bacterial growth conditions, growth stages, and even the method of preparation of the OMVs [134]. Peptidoglycans can also be found in OMVs, with approximately 0.3–0.5 ng of the muramic acid fraction of peptidoglycan per µg of protein [135].

OMVs from *H. pylori* (Hp-OMVs) have been extensively characterized. Hp-OMVs may contain genetic materials, including extracellular DNA (eDNA) and small non-coding RNAs (sncRNA). eDNA is implicated in communication between bacteria and has been detected on the surface of OMVs from the strain NCTC 11,639 of *H. pylori* [136,137]. Sequencing analyses demonstrate the presence of regulatory sncRNAs (sR-2509025 and sR-989262) in microvesicles isolated from the strain J99 of *H. pylori* [138]. These molecules can act as pathogen-associated molecular patterns (PAMPs) and interact with the pattern recognition receptors (PRRs) of target cells [139,140,141]

OMVs can perform functions, such as serving as vectors for pathogen–pathogen and pathogen–host cell communication and exchanging genetic materials and chemical compounds, signaling molecules, and virulence factors [142]. Communication can occur between the same or different bacterial strains, providing physiological advantages for bacteria, as they supply genes for antibiotic and bacteriophage resistance [143]. One important point to mention is that OMVs carry virulence factors that favor the construction of bacterial biofilms [144]; thus, these microvesicles play a key role in the survival of the bacteria in the host. They facilitate the action of bacterial virulence factors, such as toxins, directly on the epithelial cells of the host at colonization sites [142]. In the case of *H. pylori*, one of the virulence factors present in the OMVs is urease which besides alkalinizing the gastric medium also displays neurotoxic properties that potentially could contribute to AD [6].

Proteomic studies indicated that the strains of *H. pylori* 26,695 and CCUG 17,875 possess the subunit of urease (UreA) within the microvesicles. These findings corroborate the fact that AGS gastric cells treated with Hp-OMVs contained UreA in the cytoplasm and cell nucleus [145,146]. These findings indicate that OMVs can enter cells and release their contents into the cytoplasm, leading to morphological changes and cellular toxicity [146].

In addition to entering cells, OMVs can induce damage to cell junctions and transmigrate through gastric epithelial monolayers, causing an immunomodulatory effect through the deposition of antigens and virulence factors in the bloodstream [147,148]. For example, in cultures of human peripheral blood mononuclear cells, an increase in the expression of pro-inflammatory cytokines IL-6 and anti-inflammatory cytokines IL-10 was observed; in eosinophils, a degranulation process occured, releasing cytotoxic granular proteins, such as the eosinophilic cationic protein, leading to tissue degradation [147,149].

Studies indicate that catalase is present at seven times higher levels in OMVs than in the microorganism of origin. These findings suggest that OMVs possess a strong antioxidant system against the immune system [150]. LPSs are another virulence factor present in OMVs that can modulate immune response pathways, activating, for example, innate immune cells (macrophages/dendritic cells/microglia) and adaptive immune cells (T cells/B cells) in remote organs [142].

Little is known about the mechanism of action of OMVs from *P. mirabilis*, but they likely act alongside essential factors for the construction of the bacterial biofilm by carrying adhesion proteins, quorum-sensing molecules, LPSs, efflux pumps, urease, and other virulence factors [151].

#### 5.1.1. Indications of the Contribution of OMVs in Gastric and Central Nervous System Pathologies

It is interesting to highlight that extracellular vesicles from human stem cells can be used as a treatment for diseases such as Alzheimer’s and epilepsy. However, an antagonistic effect can be expected for the OMVs from pathogens [152].

The proteins present on the surface of OMVs can act as pathogen-associated patterns PAMPs and interact with the pattern recognition receptors (PRRs) of target cells [139,141,153]. This indicates that OMVs can induce a series of immune and pro-inflammatory responses in the host. When epithelial cells were exposed to OMVs, an increase in IL-8 expression and the activation of NOD1-dependent NF-κB was observed [134,135,154,155,156,157]. Microglial cells may respond through IL-1β, TNF-α, and IL-6 [142]. The production of pro-inflammatory cytokines adds to other events, such as the activation of caspase-9, caspase-8, and caspase-3; increased oxidative stress; and changes in the gene expression of molecules that affect cell morphology, culminating in host cell apoptosis. These findings lead to damage to cell junctions and/or cells of the intestinal epithelial barrier, gastric in the case of *H. pylori* infection, allowing the extravasation of OMVs and other virulence factors across the epithelial barriers [142,145,158].

Once the microvesicles breach the initial site of infection, they can migrate through the bloodstream or even via the vagus nerve to other tissues/organs of the host. The BBB is composed of astrocytes and endothelial cells, which serve to protect the central nervous system [142]. These cells have a low uptake of extracellular substances and high-quantity of molecules associated with cell junctions, strictly restricting cellular permeability [142]. Thus, only hydrophobic molecules with a mass of less than 400 daltons can permeate this structure [142].

However, pathological processes such as strokes, chronic low-grade inflammatory processes, and intestinal dysbiosis are associated with changes in BBB permeability, allowing the passage of hydrophobic molecules and hydrophilic molecules with greater molecular mass [142]. In this context, the physicochemical characteristics of OMVs allow for the permeabilization of the BBB, and their contents may contribute to BBB disruption. The presence of LPSs in OMVs (and *H. pylori* urease in Hp-OMVs) potentially permeabilizes the BBB by reducing ZO-1, claudin-5, claudin-4, and cadherin [159,160,161,162,163].

Once in the brain, OMVs are internalized by astrocytes, which induce the activation of glial cells and neuronal dysfunction, resulting in prodromal patterns of diseases such as Alzheimer’s and Parkinson’s, and they are strongly associated with the formation of Aβ-plaques and subsequent cognitive decline. A critical role of complement C3 signaling and its receptor C3a (C3aR) in mediating the interaction between astrocytes, microglia, and neurons in the presence of OMVs from *H. pylori* has also been identified [13].

OMVs lead to an increase in Aβ -plaque deposition by reducing the microglial phagocytosis of Aβ, leading to neurodegeneration and synaptic deficits [13]. The OMVs from *H. pylori* have the ability to increase interactions between astrocytes and microglia via C3-C3aR, and the microglial activation triggers neuronal dysfunction and Aβ deposition, resulting in cognitive impairment. The abnormal activation of microglia may have resulted in a reduction in the number of neurons in the CA3 region of the hippocampus in WT mice treated with Hp-OMVs. In the App NL-G-F mouse animal model, the OMVs caused cognitive impairment and memory loss through the inhibition of the C3-C3aR signaling pathway [13]. This pathway is of utmost importance when discussing immunological regulation in the CNS, mediating glial reactivity, aiding in the response to molecules that may compromise neuronal homeostasis, and ultimately influencing network function and Aβ pathology, possibly exacerbated by *H. pylori* [164,165].

Palacios et al. (2023) demonstrated in vivo that mice treated with systemic (tail vein injection) and oral administrations of *H. pylori* OMVs exhibited astrocytic reactivity, triggering the retraction of neuronal processes and inhibiting neurite growth [166]. Palacios et al. (2023) also observed that astrocytes treated with Hp-OMVs exhibit activation of NF-κB and secretion of IFNγ, with impaired neuronal dynamics [166]. These astrocytes acquired an A1 reactive profile, which is directly linked to neurotoxicity. This cellular morphology correlates with NDs such as Alzheimer’s [167]. In this configuration, astrocytes undergo increased expression of intermediate filament proteins (GFAP and vimentin), hemichannels (Cx43), membrane proteins (αVβ3 integrin), and IFNγ, which in turn impair the dynamics of neurons, resulting in a state of neurotoxicity [166].

According to these findings, Kandpal et al. (2024) demonstrated in vitro that Hp-OMVs may be promoting the activation of STAT3 and inducing the expression of signature markers associated with AD, compromising neuronal functions. This activation of STAT3 is consequent to the dimerization of STAT3 molecules and subsequent nuclear translocation after phosphorylation. Treatment with *H. pylori* secretome increased the presence of pSTAT3 in the nuclear compartment of cells, resulting in changes in protein expression [168].

Kandpal et al. (2024) also showed that the secretome of *H. pylori* induces a remarkable increase in the protein expression of APP, APOE4, PSEN1, and BACE1 in neuronal cells, indicating that the activation of STATs caused by the Hp-secretome is connected to the markers of AD [168]. In vitro and in vivo findings indicated that Hp-OMVs may induce the expression of inflammatory cytokines in microglia and decrease the viability of neuroblastoma cells, while in mice, they can lead to the presence of amyloid plaques, as well as activated microglia and astrocytes [142].

Thus, through the activation of a pro-inflammatory response, reactive microglia and astrocytes, and the disruption of cell junctions, the bacterial microvesicles may establish a relationship between bacteria and ND, exacerbating prodromal phases.

#### 5.1.2. OMVs as Therapeutic Tools 

*H. pylori* causes a resistant and persistent infection, to which OMVs may represent a new therapeutic alternative in combating bacterial infections. The importance of microvesicles in the successful colonization of the pathogen is well characterized in the literature. To allow the use of OMVs as a potential therapeutic tool, it is necessary to establish a reliable and reproducible protocol for their isolation, with an adequate choice of the growth medium, thus allowing for the artificial selection of the compounds stored in the OMVs [140,169].

In this context, it has been reported that the LPS contained in the vesicles is important for the induction of a pro-inflammatory immune response by activating immune cells such as monocytes and macrophages, in addition to triggering a humoral response. These properties enable the LPSs to direct an adaptive immune response, which allows for the conceptualization of microvesicles as carriers of epitopes for the development of attenuated or inactivated vaccines toward microorganisms [170]. However, LPSs are also a limiting factor for the use of OMVs in vaccine construction, as their excessive toxicity can lead to septic shock [130].

Probiotic bacteria can be used as a complementary treatment for the eradication of *H. pylori*, due to the production of antibacterial substances, while competing with *H. pylori* for adhesion receptors, and stabilizing the intestinal mucosal barriers [171]. Through genetic engineering and leveraging the capacity of OMVs to carry antigens of interest, including genes or inhibitors within them, it would be possible to fight a target organism or induce a memory response, as mentioned above. In the case of *H. pylori*, an example of this development would be the use of an antigen for VacA within the lumen of OMVs; however, both VacA and CagA, as well as HtrA (serine protease), can be limiting factors, similarly to LPSs, posing risks due to their carcinogenic properties [149,172,173].

An ideal scenario would involve the development of genetically modified strains without these factors, with them deleted or attenuated. Thus, Tobias et al. (2017) designed a non-toxigenic strain of *Vibrio cholerae*, modified to express adhesin A of *H. pylori* (HpaA). Oral immunization tests in mice using the mutant organism induced high anti-HpaA responses in serum, demonstrating how OMVs could work for this technology [174]. However, the lack of virulence factors has the disadvantage of potentially reducing or causing total loss of the immunogenicity attributed to them, resulting in a suboptimal immune response [130].

These findings lead us to believe that a targeted investigation into the virulence factors of microorganisms may be enlightening. Factors such as urease, LPS, VacA, and CagA appear to have significant relevance for the progression of ND, as discussed earlier in this review. Therefore, the development of genetically modified bacteria yielding non-toxigenic strains would allow the use of microvesicles as drug carriers. As an example, in the case of urease, strains could be constructed to express inhibitory genes capable of inactivating the enzyme. Through this approach, biofilm formation that allows pathogen survival and the spread of other virulence factors could be avoided, leading to a reduction in the toxicity of the organisms [175,176,177]. Therefore, OMVs present immense biological potential as immunomodulatory or antibacterial tools.

### 5.2. Lipopolysaccharides (LPSs)

LPSs are virulence factors produced by Gram-negative bacteria, consisting of glycolipids present in the outer membrane of bacteria, and found on the surface of released microvesicles [178]. LPSs comprise a highly conserved lipophilic region called lipid A, composed of fatty acids and responsible for anchoring the structure in the outer membrane [179]. Lipid-A is linked by 3-deoxy-D-manno-oct-2-ulosonic acid to a hydrophilic region called the core [180]. The core can be fragmented into internal and external domains, both regions being composed of a highly variable chain of sugars [180]. This region allows binding to the terminal portion of LPSs, which is highly variable and called the O-antigen, consisting of repeated molecules of oligosaccharides. This region engages in direct contact with the host cell during infection, has antigenic properties, and provides protection against lysozymes and antibiotics [181].

The main source of LPSs, or endotoxins, in the human body comes from intestinal microorganisms (~1 g). Low concentrations of LPS can trigger an inflammatory response in microglia through the activation of the TLR4 receptor and its co-receptor MD2 (myeloid differentiation factor 2), which recognize molecular patterns associated with pathogens. The activation of these receptors leads to the activation of NF-κβ, which induces the transcription of several inflammatory genes, including TNF-α, IL-6, and IL-1β (Figure 4) [182].

Once activated, microglia can induce astrocyte reactivity, exacerbating the inflammatory response. This activation triggers multiple pro-inflammatory pathways, including interleukins (IL-1, IL-6, IL-12, IL-17, IL-18, p40, IL-1β, and TNF-α), cytokines (CCL2, CCL5, and CXCL8), and complement system proteins (C3, C3a, and C5a receptor). These factors contribute to the increased production of ROS and nitric oxide synthase (NOS) [183]. In addition to inducing a pro-inflammatory response, LPSs can lead to leukocyte infiltration in the brain. Furthermore, the positive regulation of CD14-dependent TLR4 activation in enterocytes/macrophages results in increased intestinal permeability due to damage to the epithelial barrier [184]. The endotoxin can also activate caspases (4, 5, and 1), induce cell death by pyroptosis (pro-inflammatory cell death), or trigger microglial phagocytosis of stressed cells through the P2Y6 receptor; and activate the complement system and complement receptor 3, leading to an increased M1 inflammatory response in macrophages [185,186,187,188,189].

The involvement of endotoxin (LPSs) with NDs is still not well understood. However, studies indicate that patients with AD and PD present elevated levels of LPSs in the plasma 3–6 times more than healthy patients [7,190,191,192]. Considering that the highest production of LPSs occurs in the intestine, the increase in systemic LPS may be due to changes in intestinal epithelial permeability caused by the endotoxin itself. Other studies demonstrate that a single systemic administration of LPSs (5 mg/kg) can be used to generate a Parkinson’s model and can also induce cognitive decline in Alzheimer’s models [193,194]. In addition to compromising the BBB and promoting a pro-inflammatory response that exacerbates neurodegeneration, LPS can also contribute to the formation of neurofibrillary tangles, enhance Aβ influx into the CNS, impair its clearance, and induce cell death through COX-2 and ERK kinase pathways [183]. Possibly, these alterations explain the correlations observed in the literature, such as the presence of endotoxin and mild cognitive impairment, memory dysfunctions, and elevated levels of Aβ 1–42 and Tau [191,192].

In vivo assays indicate that LPSs can increase the expression of ApoE in mice, subsequently facilitating the clearance of endotoxin from the blood [195]. The presence of ApoE4 variants, both in humans and mice, has proven to be more detrimental following endotoxin administration. It is believed that the presence of ApoE4 increases LPS toxicity or hinders LPS clearance [196,197]. One hypothesis is that ApoE can lead Gram-negative bacteria to death; its variant ApoE4, in turn, leads to bacterial death less efficiently, which may boost infections by these bacteria, resulting in higher LPS levels [198].

Intraperitoneal injections of LPS can induce a pathology with characteristics of AD and PD in mice. The administrations of the toxin triggers upregulation of the mRNA of APP and consequently an increase in Aβ production through decreased α-secretase activity while enhancing the activity of BACE-1 and β- and λ-secretase. LPSs can also reduce the clearance of Aβ (decreasing the expression of low-density lipoprotein receptor-related protein 1, inhibiting the entry of Aβ into brain blood vessels, and resulting in the dysfunction of P-glycoprotein), which is associated with the formation of diffuse plaques, and directly promote Aβ aggregation [199,200,201,202,203,204,205].

The fact that Aβ itself has an antimicrobial role suggests that brain infections can induce the production and aggregation of β-amyloid, resulting in increased inflammation and phagocytosis, subsequently generating a degenerative effect and, consequently, cognitive deficits [198,206,207].

In addition to the effects that are directly related to AD, the intraperitoneal administration of endotoxin can also lead to the rupture of intercellular junctions, causing the detachment of endothelial cells and increasing the permeability of the BBB, activating microglia, inducing apoptosis of neuronal cells, and increasing pro-inflammatory interleukins (TNF-α, IL-1β, IL-6, IL-8, and IL-10) [204].

One of the main characteristics of AD is the presence of the phosphorylated Tau protein. Since LPSs induce microglial activation, it can release interleukins responsible for increasing neuroinflammation and it is capable of activating kinases (cyclin-dependent kinase 5 [CDK-5], GSK-3β, mitogen-activated protein kinase [MAPK], c-Jun, and p38) and hyperphosphorylating the Tau protein, leading to its aggregation [198,208]. Moreover, the fact that the endotoxin can permeabilize the BBB results in an increased dissemination of Tau proteins in the brain [209].

Several studies have demonstrated the presence of LPSs in the brain. However, there is no consensus on how the endotoxin is capable of permeabilizing the BBB or permeating through it. Various proposals on how LPSs can pass through the BBB are described in the literature [210,211,212]. One possibility is the increased pro-inflammatory response that can damage the cell junctions of the barrier, leading to a higher permeability [213]. The enhanced pro-inflammatory response decreases the expression of adhesion molecules in endothelial cells (P-selectin, intercellular adhesion molecules-1, and vascular cell adhesion molecules-1), allowing LPSs to enter the BBB through the recruitment of peripheral immune cells since LPSs can be linked to CD14/TLR4 cells or be phagocytosed by other immune cells [184,214,215]. LPSs can also pass through the barrier bound to mediating proteins, such as lipopolysaccharide-binding proteins (LBPs) and crossing the BBB using class B type I scavenger receptors, apolipoprotein AI, and ApoE [184,214]. Finally, the endotoxin can reach the brain directly through the presence of Gram-negative bacteria or with the assistance of OMVs [216].

The activation of microglia through LPS administration appears to influence various routes of diseases related to ND. The endotoxin upregulates the expression of NADPH oxidase 2 in microglia, leading to the increased production of ROS [190]. The intraperitoneal administration of LPS increased phagocytosis in microglia, as well as COX-2 and enzyme iNOS levels, directly affecting brain synapses and leading to impairments in memory, cognition, and motor function [217,218]. The intraperitoneal administration of LPS in mice induced neuronal loss in the hippocampus and prefrontal cortex and decreased the cholinergic innervation of the parietal cortex and even induced in vitro damage to myelin through the degradation of the brain’s basic myelin protein [194,219].

Moreover, the increased ROS (superoxide [O_2_^−^] and hydrogen peroxide [H_2_O_2_]) caused by the endotoxin may react with iron through the Fenton reaction and generate hydroxyl radicals (OH^−^). Antioxidant molecules are reduced in animal models treated with LPSs. The excessive presence of these reactive molecules leads to cell death through the intrinsic apoptotic pathway (mitochondrial). LPS increases the expression of neuronal TLR4 and binds to this receptor, inducing the transcription of caspase-11, which promotes the activation of inflammasomes [193,220,221,222]. The endotoxin can also directly reduce the activity of mitochondrial complexes I, III, and V of the electron transport chain, promoting greater production of superoxides, which directly affects dopaminergic neurons [193,223].

Choi et al. (2018) reported that the oral administration of *P. mirabilis* led mice to develop motor and biochemical symptoms characteristic of PD. In this same study, the authors suggested that the observed increases in α-synuclein aggregation and dopaminergic neuronal death were due to the LPSs derived from *P. mirabilis* [46]. Other studies have demonstrated that the intraperitoneal administration of the endotoxin can generate symptoms characteristic of PD in both rats and mice [188].

The effects of endotoxin on PD are associated with the increased permeability of the gastric epithelium and of the BBB, as described for AD. One hypothesis for the onset of the disease is the ascent of intestinal pathology to the brain given that aggregated α-synuclein has been found in the colon before its presence is observed in the brain. These findings correlate with the high concentration of Gram-negative bacteria in the colon, which, in turn, whose LPSs can activate the expression of the *SNCA* gene, thus increasing the expression of α-synuclein that after phosphorylation, is prone to aggregation [224]. Endotoxin may favor the deposition of α-synuclein in the brain by increasing endothelial permeability, allowing the migration of both molecules.

Furthermore, the endotoxin also leads to the activation of the *LRRK2* gene, which enhances dopaminergic neuronal death [225]. The administration of endotoxin intraperitoneally or via stereotaxic route results in microglial activation and the subsequent death of noradrenergic neurons in the locus coeruleus and dopaminergic neurons in the substantia nigra [226,227,228]. This same effect has been observed in mixed neuron–glia cell cultures [229].

The administration of LPS via intranigral injection results in a reduction in the time spent in the open arm in the elevated plus-maze test, indicating depressive and anxious-like behavior (symptoms that precede PD), likely caused by the reduction in dopaminergic and noradrenergic neurons and brain-derived neurotrophic factor (BDNF), which is important for synaptic plasticity and memory processes [230].

### 5.3. Ureases

Ureases are virulence factors necessary for the survival of microorganisms. These nickel-dependent enzymes catalytically hydrolyze urea, producing ammonia and carbamate, which subsequently decomposes into carbon dioxide and another molecule of ammonia. The production of ammonia alkalinizes the surrounding medium and precipitates salts that facilitate the survival and proliferation of the pathogen [231].

Ureases share approximately 55% of their peptide composition across various kingdoms, including plants, fungi, archaea, and bacteria. The structure of the functional monomer of ureases can be composed of different subunits: α in plants (ex: *Canavalia ensiformis*) and fungi; α-β (ex: *H. pylori, Yersinia pseudotuberculosis*), and α-β-λ in most bacteria (ex: *Proteus mirabilis*, *Klebsiella* sp.). However, the conformation of the tertiary structure is conserved regardless of origin. Varying only in their quaternary structures, bacterial ureases are mostly trimers of the functional monomers, possessing α-β-λ subunits, i.e., [α, β, γ]_3_. An exception to this is the urease from *H. pylori*, which consists of a dodecameric quaternary structure with α and β subunits, i.e., (α,β)_6_ [232]. The sharing of peptide sequence and tertiary structures allows various non-enzymatic effects to be conserved also among different ureases [233].

Our group has studied the non-enzymatic effects of urease for over four decades. In 1981, Carlini and Guimaraes isolated an isoform of urease from *C. ensiformis*, called canatoxin (CNTX), endowed with neurotoxic properties, causing seizures that precede the death of rodents [234,235]. Similar effects were observed with Jack Bean Urease (JBU), the major isoform of urease found in *C. ensiformis* [236,237].

Studies have shown that CNTX causes an array of non-enzymatic effects on in vivo models, such as hypothermia; bradycardia; hypotension; hypoxia; hyperinsulinemia; hypoglycemia; and inflammation, such as paw edema in rats [234,238,239,240]. The non-enzymatic effects of CNTX and JBU, observed even in the absence of ureolytic activity, were also seen in vitro, as the ureases were able to induce the exocytosis of neurotransmitters, elevate intracellular calcium levels, and interact with lipid membranes, forming cation-selective ion channels [237,240,241,242,243].

The nervous system of insects has served as a model for studying the neurotoxic effects of ureases [244]. JBU induces behavioral changes and alterations of the heart rate frequency in the cockroach *Nauphoeta cinerea*, acting on the cholinergic, octopaminergic, and GABAergic systems of the insect’s central and peripheral nervous systems [245]. Carrazoni et al. (2016) also demonstrated through electrophysiological recordings that JBU induced neuromuscular blockades by reducing muscle contraction tension and decreasing action potential frequencies. The same study also showed that JBU can induce the release of neurotransmitters in different insect models. Considering the ability of JBU to increase Ca^2+^ influx, it is possible that JBU may also interact with K^+^ channels as part of the mechanism underlying the exocytosis of neurotransmitters [246].

Since cytokine production can compromise the integrity of the BBB, the production of a peripheral inflammatory response becomes relevant in the scenario of ND. The peripheral pro-inflammatory effect induced by ureases has already been well characterized by our group. Two hours after the intraplantar injection of CNTX, an increase in paw edema in rats was observed, indicating a chemotactic (monocytes, macrophages, neutrophils, and mononuclear cells) and inflammatory effect at the site [240]. The edema is mediated by the degranulation of mast cells and an increase in the release of histamine and serotonin [247,248]. Inflammation promoted by CNTX was also observed in the air-pouch model in rats, dependent on the urease’s chemotactic effect on macrophages [241].

A similar effect of inducing paw edema was observed for *H. pylori* urease (HPU) in mice. The pre-treatment of mice with dexamethasone or esculetin (inhibitors of phospholipase A2 and lipoxygenase, respectively) reduced the edema, indicating mediation by the eicosanoid pathway in this process, particularly through the synthesis of eicosanoids [249]. Boyden chamber assays indicated that HPU, at nanomolar concentrations, acts as a potent chemotactic factor to attract human neutrophils. Neutrophils activated by HPU had increased ROS production and inhibition of apoptosis due to the augmented expression of Bcl-XL and decreased levels of Bad, causing active cells to survive longer and have a sustained production of pro-inflammatory molecules [249].

Moreover, HPU can also induce the expression of iNOS and increase NO production in macrophages [250]. It stimulates the production of IL-1β, IL-6, IL-8, and TNF-α in monocytes; IFN-γ, IL-12p40, and IL-10 in mononuclear cells; and IL-1β in AGS cells (gastric epithelium) [251,252]. Urease not only aids in the pathogen’s survival in the gastric environment but also becomes internalized by gastric epithelial cells, leading to the expression of angiogenic mediators [253]. Assays in gastric cell cultures (MKN28) treated with *H. pylori* revealed damage to cell junctions [254]. Souza et al. (2019) treated the human microvascular endothelial cell line (HMEC-1) directly with HPU and observed changes in the cytoskeleton, accompanied by increased ROS, NO, and IL-1β, making the cells more permeable due to damage to cell junctions mediated by VE-cadherin phosphorylation [255]. Assays with the urease-negative mutants of *H. pylori* indicated the loss of the ability to induce damage to cell junctions [254].

The interaction between the pathogen and host activates inflammasomes, leading to the activation of caspase-1 and the release of IL-1β and IL-18. Assays with ΔHPU *H. pylori* mutants in mouse and human dendritic cell cultures have shown that in the absence of HPU, the assembly of the NLRP3 inflammasome is compromised, and the activation of caspase-1 and secretion of IL-β do not occur [256]. HPU binds to the CD74 receptors of gastric epithelial cells, inducing the activation of NF-κB and the production of IL-8 and exacerbating the inflammatory response [257].

Electrophysiological assays demonstrate that long-term potentiation (LTP) is inhibited by ureases (JBU and CNTX) in mouse and rat hippocampal slice models, respectively. In hippocampal slices, CNTX was able to induce long-term depression (LTD) [237]. These ureases promoted the exocytosis of neurotransmitters such as serotonin and glutamate [237,258]. It is likely that these alterations are linked to the ability of ureases to induce an increase in Ca^2+^ influx, as demonstrated in platelets [231] and in primary hippocampal cultures from rats.

Due to the homology among the sequences and conserved tertiary structures, it is expected that the biological effects of ureases will also be highly conserved. Intraperitoneal administrations of HPU have been shown to produce hypothermia, seizures, and death, similarly to findings with CNTX [259]. Furthermore, the administration of HPU (5 μg/animal/day) intraperitoneally for 7 days induced biochemical changes in the rat brain, resembling the prodromal phase of AD, such as the increased expression of Iba1 (marker of activated microglia) and the hyperphosphorylation of tau at Ser199, Thr205, and Ser396; however, no change in the total amount of tau or GSK-3β was observed (Figure 4) [6].

When HPU is incubated with SH-SY5Y cell lines (human neuroblastoma), an increase in the production of ROS and elevated intracellular calcium levels were seen, and similar effects on the same cell line were observed for the urease of *P. mirabilis* (PMU). In BV-2 cells (murine microglia), HPU is capable of inducing the production of ROS, IL-1β, and TNF-α and decreasing cell viability. In turn, PMU does not alter the cell viability of BV-2 but induces an increase in the production of IL-1β and TNF-α [6,260].

It is important to highlight that PMU is also targets the interior of the nucleus and possibly acts as a transcription agent, leading to alterations in the expression of other proteins [260]. Recent work from our group has demonstrated that intraperitoneal treatment with PMU (20 μg/animal/day) induced depressive-like behavior in mice, while biochemical analyses showed that the animals treated with PMU had decreased levels of tyrosine hydroxylase (a marker of dopaminergic neurons) and of soluble α-synuclein in brain extracts [261]. In vivo treatments with PMU also induced hyperglycemia in mice. In vitro analyses showed that PMU increased IL-1β and TNF-α levels in Caco2 cells (colorectal adenocarcinoma lineage), increased paracellular permeability in Hek 293 cells (renal endothelium), and caused alterations in the process of α-synuclein fibrillation, which resulted in amorphous aggregates (a similar effect was observed for HPU) [261].

It is noteworthy that, like LPS, ureases have similar mechanisms for introducing themselves into the brain. Ureases can reach the brain through peripheral damage caused by their pro-inflammatory activity, with reflexes on the intestinal cell junctions and the BBB [232]. This allows for increased permeability at cell junctions, enabling urease, which is predominantly present in the intestine as a product of the gut microbiota, to migrate into the bloodstream and subsequently access the CNS. Another possibility is its transportation within OMVs that can cross the BBB or migration via the vagus nerve, or it can be carried internalized in immune cells [232].

### 5.4. Vacuolating Protein A (VacA) and Cytotoxin Associated with Gene A (CagA)

Both virulence factors are associated with a successful infection by *H. pylori*, as VacA and CagA contribute to the death of gastric epithelial cells, facilitating the migration of pathogen beyond intestinal boundaries. Infections with VacA/CagA-positive strains are associated with intense neutrophil infiltration and a strong inflammatory response mediated by ROS and interleukins (IL-1 α, IL-1 β, IL-2, IL-8, and TNF α), extensive DNA damage in epithelial cells, and morphological alterations (Figure 4) [262,263].

The VacA protein is capable of generating vacuoles in the cellular cytoplasm, causing mitochondrial damage and disrupting cell–cell junctions and thus increasing epithelial permeability at the expense of cell death [263]. In turn, the virulence factor CagA is delivered to epithelial cells via a type IV secretion system; subsequently, it is phosphorylated by tyrosine kinase. When phosphorylated, CagA can activate various intracellular signaling transduction pathways that result in tissue inflammation and the rearrangement of the cellular cytoskeleton [262].

The formation of vacuoles occurs after the oligomers of VacA are internalized by adjacent cells, and once in the cytoplasm, the virulence factor induces the formation of vacuoles with characteristics of late endosomes and early lysosomes [263]. When oligomerized, VacA can form anionic membrane channels that facilitate the transport of chloride ions [264].

In addition to inducing vacuolization, once present in the cytoplasm, the virulence factor can interact with the mitochondria. When VacA is internalized, it targets the mitochondrial membrane and induces the dysfunction of the organelle, with increased permeability of the outer mitochondrial membrane, dissipating the mitochondrial transmembrane potential and leading the cell to a positive regulation of pro-apoptotic proteins (Bax and Bak) and negative regulation of the anti-apoptotic protein Bcl-2 [265,266,267]. The formation of membrane pores allows for the release of cytochrome c into the cytoplasm, which leads to the activation of executioner caspases, resulting in cell death. Another possibility for cell death caused by VacA is through the activation of dynamin-related protein 1 (Drp1), which inhibits mitochondrial fission, resulting in cell death [268].

Furthermore, VacA is capable of activating p38 MAPK, leading to the activation of transcription factor activator protein 2 and NF-κB and positively regulating pro-inflammatory interleukins and iNOS [269]. Studies suggested that this pathway is related to microglial and astrocytic inflammation caused by the deposition of Aβ (1–42) and cholinergic hypofunction, resulting in cognitive impairment [270,271].

On the other hand, CagA negatively regulates autophagy/apoptosis through the positive regulation of Bcl-2 and Bcl-XL while it activates c-Met-PI3K/Akt-mTOR signaling, leading to an increased pro-inflammatory response mediated by IL-8, IL-1β, and TNF-α [272]. Brandt et al. (2005) demonstrated that the increase in interleukins can also be caused via Ras → Raf → Mek → Erk → NF-κB [273]. The survival of endothelial cells due to the negative regulation of autophagy/apoptosis caused by CagA allows them to remain alive but stressed, while activated pro-inflammatory signaling resultant from the increased reactive species causes cell death due to DNA damage, elevating intestinal permeability [274].

Studies in humans with CagA-positive *H. pylori* strains have shown variations in dopamine and serotonin levels in the brain [275].

### 5.5. Hemolysins and Fimbriae

Hemolysins are pore-forming toxins found on the cytoplasmic membrane of pathogens that induce the lysis of red blood cells [276]. Additionally, the pore structure of hemolysins acts similarly to amyloid proteins to induce membrane permeabilization [277]. However, effects independent of cell lysis are observed with hemolysins, such as neurodegeneration, neuroinflammation, and motor impairment [276].

In vivo studies demonstrated that the hemolysin from *Staphylococcus aureus* promotes the destruction of plasma membranes in nerve cells, increasing the extracellular levels of amino acids such as glutamate, aspartate, and GABA, causing the death of glutamatergic and GABAergic neurons [278]. The hemolysin from *P. mirabilis* leads to the aggregation of α-synuclein through mTOR-p mediated signals, while the formation of pores in the plasma membrane accelerates the aggregation of intestinal α-synuclein [276]. On the other hand, the hemolysin from *S. aureus* can activate the NLRP3 inflammasome, inducing the release of IL-1β and subsequent programmed cell death (necrosis) [279].

Fimbriae are long polymeric filaments located on the surface of bacteria that aid in adhesion to surfaces, facilitating colonization. They are produced by both Gram-negative and Gram-positive bacteria [280]. Fimbriae can be directly associated with the efficiency of pathogens, allowing for interactions with host cell receptors and eliciting a pro-inflammatory response. Once internalized by host cells, fimbriae facilitate the migration of the pathogen and of its virulence factors [281,282]. The fimbriae of *Porphyromonas gingivalis* facilitate the infection of dendritic cells by the pathogen, inducing the cells to undergo a maturation process with the subsequent production of IL-1β, IL-6, TNF-α, IL-10, IL-12, and IFN-γ (Figure 4) [283].

### 5.6. Catalase

Catalase is an enzyme that catalyzes the breakage of hydrogen peroxide into water and oxygen, aiding microorganisms to mitigate the effects of ROS. Aβ is capable of binding to catalase, thereby reducing its enzymatic activity, leading to increased hydrogen peroxide levels and the conditions of oxidative stress characteristic of AD [284].

Unlike Aβ, α-synuclein interacts with catalase differently. The inhibition of catalase promoted by α-synuclein occurs due to the suppression of gene expression. α-synuclein disrupts the activity of transcription factors (PPARγ), preventing catalase expression. The absence of catalase allows for increased hydrogen peroxide levels, which are associated with PD [285].

### 5.7. Short-Chain Fatty Acids (SCFA)

Short-chain fatty acids (SCFA) are molecules composed of monocarboxylic acids with up to six carbon atoms. These molecules are produced through the anaerobic fermentation of indigestible polysaccharides (e.g., dietary fibers). SCFAs consist primarily of acetate, propionate, and butyrate. They possess neuroprotective properties by aiding in mucus production, regulating neurotransmitter generation, reducing local inflammation, and contributing to the maintenance of the integrity of the intestines and BBB [286].

Meta-analysis studies across different countries have shown that the gut microbiota in PD patients exhibits an increase in microorganisms associated with mucin degradation (e.g., *Akkermansia*) and a decrease in SCFA-producing microorganisms (e.g., *Roseburia*, *Fusicatenibacter*, *Faecalibacterium*, *Lachnospiraceae ND3007*). This condition results in lower levels of acetate, butyrate, and propionate. Moreover, the reduction in SCFA-producing organisms is associated with decreased biotin metabolism, glycan degradation, and reduced biosynthesis of amino acids (phenylalanine, tyrosine, and tryptophan) [266].

Butyric acid, produced by these microorganisms, is linked to mucus production in the intestinal epithelium. This molecule plays a protective role in maintaining cellular junctions and inhibits the NF-κB signaling pathway. The absence of butyrate compromises intestinal permeability in PD patients. Additionally, butyrate has anti-inflammatory properties that operate through various mechanisms: (i) It induces regulatory T-cell production and *FOXP3* expression, leading to histone deacetylase inhibition. (ii) It acts directly on dendritic cells and macrophages via interactions with GPR109a (a G-protein-coupled receptor), promoting the differentiation and expansion of Treg cells and IL-10-producing T cells. When associated with this receptor, butyrate can also suppress inflammation by inhibiting IL-18 production. (iii) It interacts directly with GPR41 and GPR43. The decrease in SCFA-producing bacteria in PD patients may contribute to increased neuroinflammation and, subsequently, neurodegeneration [266].

### 5.8. Amyloid Proteins

Bacterial amyloids (ABs) assist microorganisms in colonization by directly acting as part of the biofilm matrix, increasing hydrophobicity and adhesion to surfaces. Amyloid proteins are monomers capable of forming insoluble fibers through their structural features. The ability of these proteins to fold into β-sheet structures oriented perpendicularly to the fiber axis underlines their fibrillation property. These proteins are referred to as functional amyloids and provide protection against host and environmental defenses [287].

Interactions between prokaryotic and eukaryotic amyloids can occur either directly or indirectly through the immune response triggered by ABs. Several species of microorganisms can produce this class of protein, such as *E. coli* (CsgA and CsgB; curli), *Salmonella typhimurium* (curli), *Pseudomonas aeruginosa* (FapC), *Bacillus subtilis* (TasA), and *S. aureus* (PSMα) [287].

ABs act as microbe-associated molecular patterns (MAMPs) and can induce an increased immune response in the host through interactions with MHC class I and macrophage activation. Additionally, these molecules can lead to the release of ROS and NF-kβ through cytokine release mediated by TLR2, resulting in a neuroinflammatory response (Figure 4). ABs, in their monomeric or aggregated forms, may contribute to initiating the fibrillation of eukaryotic amyloids associated with ND [287].

#### 5.8.1. Phenol-Soluble Modulin

Phenol-soluble modulins (PSMαs) are amphipathic peptides produced by *S. aureus*, associated with the formation of extracellular fibrils and the stabilization of bacterial biofilms, enhancing adhesion and hindering antibiotic treatment. PSMαs can induce immune responses such as extracellular traps (ETosis), which colocalize with amyloids in human tissues [288].

In vitro analyses indicate that PSMαs are capable of accelerating the fibrillation of α-synuclein. The fibrillation rate differed for each type of PSMα tested. PSMα1 and PSMα4, which have structures like amyloid fibrils, were able to induce α-synuclein aggregation at lower concentrations than PSMα2 and PSMα3, which exhibit α-helical fibrillar conformations. Furthermore, it was observed that α-synuclein aggregates with PSMα, and when incubated with HEK cells, PMSα induces α-synuclein aggregation and phosphorylation within the cells [288].

However, the effect of PSMα does not occur in the same manner for the Aβ_40_ peptide. Studies indicated that the interaction between PSMα3 monomers and Aβ_40_ induces a dose-dependent effect, delaying aggregation and altering the fiber formation process into a granular conformation (Figure 4). On the other hand, PSMα3 oligomers accelerate the fibrillation process of Aβ_40_. These differences are attributed to the ability of the compounds to form chemical bonds: the monomer interacts via hydrophobic and electrostatic processes in the Asp1-Ala2 and His13-Val36 regions of Aβ_40_, preventing conformational transitions of the eukaryotic amyloid. Meanwhile, the oligomer can interact with the C-terminal region of Aβ_40_ through weak hydrogen bonds and hydrophobic interactions, enabling conformational transitions [289].

#### 5.8.2. FapC

The amyloidogenic protein FapC plays an important role in the composition of the extracellular amyloid matrix of *Pseudomonas aeruginosa*, enabling the stabilization and survival of bacterial biofilms [290]. Due to the physicochemical properties of FapC, it can act as a catalyst for the fibrillation of Aβ in vitro and in zebrafish. The removal of highly repetitive amino acid regions in FapC (ΔR1R2R3, ~35 residues) reportedly allows the formation of disulfide bonds, which delay fibrillation and result in thinner, more fragmented fibrils of Aβ and α-synuclein [291].

The co-administration of FapC with Aβ resulted in the acceleration of all the analyzed effects in zebrafish, including eukaryotic amyloid fibrillation in the cerebroventricular space, increased ROS levels in brain homogenates, and behavioral and cognitive impairments (Figure 4) [291].

#### 5.8.3. Curli

Curli proteins are extracellular fibers associated with cell aggregation, surface adhesion, biofilm formation, and cell-to-cell adhesion. In *E. coli*, the Curli protein is composed of five subunits, CsgA, CsgB, CsgE, CsgF, and CsgG, while in *Salmonella*, Curli proteins are referred to as thin aggregative fimbriae (Tafi) [292].

These proteins share high similarity with eukaryotic amyloid fibers, being composed of β-sheets, and they are resistant to protease degradation. In silico studies indicated that Curli subunits (CsgA and CsgB) may adopt conformations analogous to eukaryotic amyloid fibers, driven by interactions between glutamine and asparagine residues, forming a stable hydrogen bond network [292].

Increased levels of Curli are directly correlated with elevated Aβ and Tau levels in the serum of patients with AD. Like Curli, homologous CsgA proteins of other microorganisms are associated with the aggregation of eukaryotic amyloids, although aggregation kinetics vary for each homolog [292].

When *E. coli*-expressing Curli is included in the diet of *Caenorhabditis elegans*, there is an increase in α-synuclein aggregation in muscle cells. Curli’s effects are also evident in other experimental models. Rats exposed to the protein can develop motor impairments, neuroinflammation, increased α-synuclein deposition in intestinal ganglion cells, and in neurons in the striatum and hippocampus. In vitro studies revealed that Curli colocalizes with α-synuclein and that the CsgA protein may be the biologically active component capable of accelerating the fibrillation of α-synuclein. When the CsgA region is excluded from the Curli protein, these neurotoxic effects are no longer observed in *C. elegans* or human neuroblastoma cells [287].

Curli can bind to extracellular matrix proteins such as fibronectin and laminin and can induce plasmin zymogen activation. Once plasminogen is activated, bacteria can invade deeper tissues facilitated by the soft tissue degradation caused by plasmin [292].

Moreover, Curli proteins can interact with MHC class I receptors and adhere to receptor-presenting cells without influencing antigen processing and presentation, effectively using them as a “Trojan horse”. Studies suggested that Curli is absorbed by human neuroblastoma cell lines. The immune system activation is triggered by pathogen-associated molecular patterns (PAMPs) present in this protein and mediated by Toll-like receptor 2 (TLR2), resulting in the release of pro-inflammatory cytokines and IL-8 and reactive astrogliosis (Figure 4) [292,293].

Other notable findings are associated with Curli proteins. Mice treated with *E. coli*-producing Curli showed a reduction in the number of cholinergic neurons; extensive demyelination of white matter in the lumbar spinal cord; increased mRNA expression of Ceclin, p62 (an autophagy marker), and Murf1; elevated inflammatory markers; and increased α-synuclein aggregation [266,293,294]

## 6. Development of Potential Treatments

The close relationship between microbiota and NDs prompts for the search of new therapeutic alternatives. Chronic infection by microorganisms may contribute to ND, and therefore, preventive treatment that is specifically designed and with high sensitivity towards the pathogens could represent a strategic alternative in combating the development/progression of these diseases. However, not only pathogen eradication but also inflammatory control and the inhibition of virulence factors produced by the organisms are important therapeutic approaches.

### 6.1. Vaccine

Prospective cohort studies have shown that vaccines against diseases like diphtheria, tetanus, polio, and influenza are associated with lower risks of AD compared to unvaccinated patients [115]. One interesting alternative to be explored is the use of the virulence factors themselves as vaccines. In this regard, a study published by Ablkar et al. 2015 explored a possible immunization scheme based on recombinant urease (rUrease) to inhibit infections caused by the bacteria *Brucella abortus* and *Brucella melitensis*. This study found that by applying different concentrations of this urease to BALB/c mice, either subcutaneously (SC) or by intraperitoneal injection (IP), immune responses were successfully mounted, with IP inducing higher IgG titers than SC immunization and SC yielding higher levels of protection against the pathogen. Furthermore, both forms of rUrease application induced humoral and cell-mediated immune responses, increasing the production of IgG1 and IgG2a. The study indicated that the SC administration of rUrease could be a peripheral route of vaccination [295]. Thus, urease may also be the key to creating a vaccine against *H. pylori*, a microorganism directly linked to AD.

Studies report, for example, in AD, the existence of many barriers to the creation of vaccines: in this case, the heterogeneity of AD, the complexity of the immune response, and the need for adequate biomarkers to monitor efficacy. Thus, there are two lines of development: first, passive vaccines that involve the administration of preformed antibodies that bind to β-amyloid, helping to remove it, like monoclonal antibodies [296]. The second line would be active vaccines that aim to induce a robust immune response against antigens associated with AD, such as β-amyloid. In this second case, some virulence factors can be included as candidates for vaccine development [296].

### 6.2. Fecal Microbiota Transplantation

The use of fecal microbiota from a healthy patient to replace the intestinal microbiome of a diseased patient is known as fecal microbiota transplantation. Due to constant discoveries regarding organism-to-organism relationships, this method has been increasingly used in clinical trials. Another factor contributing to the use of this approach is the identification of correlations between specific pathogen groups and particular diseases [107].

In light of these discoveries, the technique has shown cure rates of ~90% in patients treated with transplants to minimize the effects of severe *Clostridioides difficile* infections. It has also shown improvements in controlling a wide range of extra-intestinal diseases, including autism spectrum disorder, multiple sclerosis, and myoclonic dystonia [107].

### 6.3. Probiotics, Prebiotics, and Diet

The intestinal microbiota plays a significant role in the development of neurodegenerative diseases. Understanding the role of each microorganism in the progression of these diseases allows us to modulate the microbiota in favor of a healthy state. This modulation can occur through diet, probiotics, or prebiotics. Through modulation, it could be possible to reduce neuroinflammation, balance neurotransmitter production (as bacteria can produce neurotransmitters through their metabolism), and increase the levels of neuroprotective metabolites, such as short-chain fatty acids [297].

Antimicrobial peptides are molecules naturally produced by the host and exhibit antibacterial, antiviral, antifungal, and antiparasitic properties. These peptides represent a promising therapeutic alternative to antibiotic resistance, as they are less susceptible to resistance mechanisms. Additionally, they can differentiate between the membrane structures of the host and bacteria. Their interaction with the membrane occurs through electrostatic attraction, allowing them to induce alterations in membrane integrity, ultimately leading to cell death [298].

Polyphenols are organic compounds widely found in cereals, fruits, vegetables, coffee, and other foods. These molecules can be classified into coumarins, tannins, phenolic acids, and flavonoids. Additionally, polyphenols can act as prebiotics, as they are resistant to host digestion and interact with the intestinal microbiota [299,300].

Among the main benefits of polyphenols are their antioxidant and anti-inflammatory properties. They can also inhibit bacterial biofilm formation by interfering with the quorum-sensing mechanism, mediated by acyl-homoserine lactone [299,300].

In addition to promoting gut health, polyphenols contribute to microbiota balance by stimulating the growth of probiotic bacteria, such as *Bifidobacteriaceae* and *Lactobacillaceae*, while reducing the presence of pathogenic bacteria, such as *E. coli*, *Clostridium perfringens*, and *H. pylori* [299,300].

In the nervous system, polyphenols regulate NF-κB signaling, prevent the formation of toxic Aβ oligomers, and modulate Tau protein hyperphosphorylation, thereby reducing the formation of neurofibrillary tangles and attenuating neurodegenerative processes [299,300].

## 7. Conclusions

In this review, we explored the characteristics of dysbiosis occurring in neurodegenerative diseases and how bacterial virulence factors contribute to the worsening or development of these pathologies, and we summarized existing and potential therapeutic approaches to combat the implicated microorganisms and their contribution to these diseases. The microbiota in the digestive system or in the oral cavity produces virulence factors for protection against host defenses, many of them acting as a Trojan horse that affect immune cells and thereby contributing to neurodegenerative diseases. The disruption of the intestinal and blood–brain barriers consequent to the inflammatory conditions associated with these pathogens, which allows the entrance into the CNS of different compounds, and distinct host–pathogen interactions underline the many ways that an imbalanced microbiota may influence the pathogenesis of neurodegenerative diseases. Among the many virulence factors reviewed here are LPSs, urease, curli proteins, amyloidogenic proteins, VacA, and CagA, which can be enclosed inside bacterial outer membrane vesicles that permeate cell membranes and enable distal communication, impacting various neurodegenerative conditions, such as Alzheimer’s and Parkinson’s diseases.

## Figures and Tables

**Figure 1 molecules-30-00687-f001:**
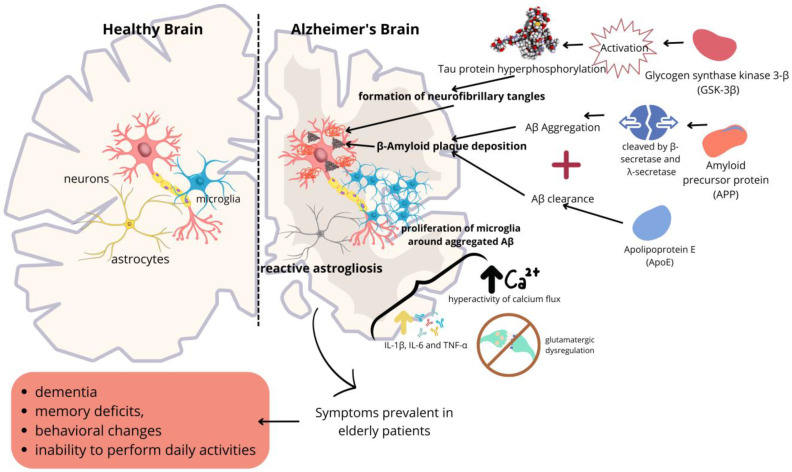
Summary of the complex pathophysiology of Alzheimer’s disease. The disease is characterized by a series of changes in the brain, such as the deposition of β-amyloid plaques and the formation of neurofibrillary tangles. The irregular unfolding of amyloid precursor proteins (APPs) and ApoeE generates the deposition of β-amyloid plaques, while a high activation of the GSK-3β enzyme promotes the hyperphosphorylation of Tau proteins, which polymerizes to form the tangles. These abnormal protein aggregates promote astrocyte reactivity, microglia activation, increased Ca^+^ influx, increased pro-inflammatory factors, decreased synapses, and neuronal death. Throughout the lives of patients, the main cognitive symptoms are dementia, memory deficit, changes in behavior, and inability to perform daily tasks.

**Figure 2 molecules-30-00687-f002:**
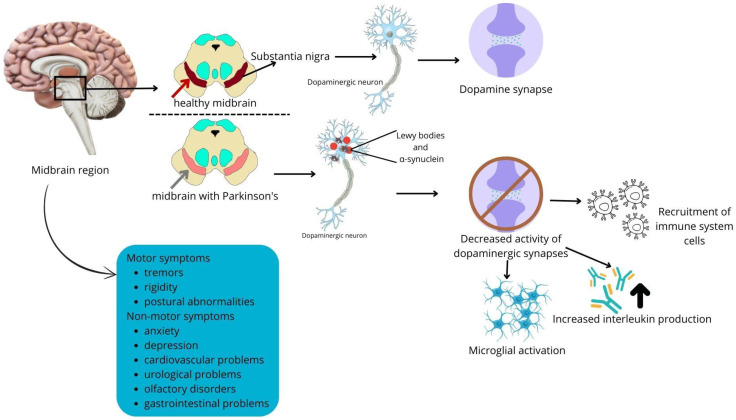
Mechanisms that lead to neuronal death in Parkinson’s Disease. In the midbrain region, more specifically in the substantia nigra, the red arrows indicate a dense region with dopaminergic neurons, while the gray arrow indicate a decrease in dopaminergic neurons due deposit of aggregated proteins such as α-synuclein forms Lewy bodies, ultimately leading to cell death and causing a decline in the number of dopaminergic neurons. Consequently, there is a decrease in the number of dopaminergic synapses and the hyperactivation of microglia and an increase in pro-inflammatory cytokines and activated immune cells. Among the symptoms, patients present motor deficits, such as tremors and rigidity, and non-motor symptoms, such as gastrointestinal problems and depression.

**Figure 3 molecules-30-00687-f003:**
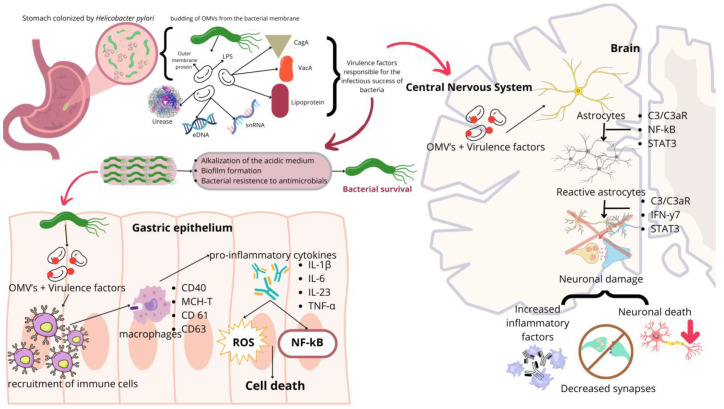
*H. pylori* OMVs and the complex ways they could interfere with neurodegenerative diseases. OMVs play a fundamental role in the formation of biofilm and microbial resistance, favoring bacteria and worsening infections. In addition, these vesicles, due to their content of virulence factors, are internalized near the junctions of gastric epithelial cells and generate an increase in pro-inflammatory cytokine IL-8, apoptosis, the rupture of gastric epithelial cell–cell junctions, vacuolization, and the formation of micronuclei. Reaching the bloodstream and through the vagus nerve, OMVs can cross the blood–brain barrier and cause a homeostatic imbalance in cells, mainly astrocytes, which acquire a pro-inflammatory profile, leading to the recruitment of immune cells, decreased synapses, and the death of neurons.

**Figure 4 molecules-30-00687-f004:**
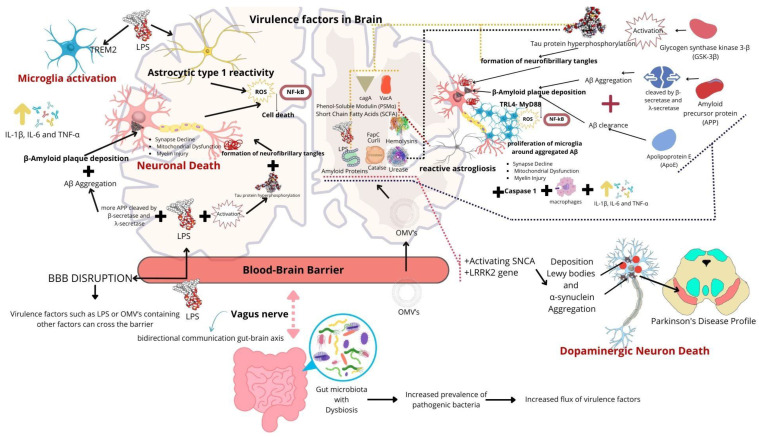
Mechanism of virulence factors in neurodegenerative diseases. The figure illustrates the mechanisms by which the LPS, a key membrane component of Gram-negative bacteria, contributes to neuroinflammation and the progression of neurodegenerative diseases such as Alzheimer’s and Parkinson’s. In addition to LPSs, other bacterial virulence factors, such as ureases, VacA, CagA, hemolysins, fimbriae, catalase, and amyloid proteins, are also involved in the pathogenesis of NDs. Ureases, VacA, and CagA can induce neurotoxic and pro-inflammatory effects by inducing hyperphosphorylation of tau protein, indicated by the black and yellow dashed line. Meanwhile, bacterial amyloid proteins can accelerate the aggregation of eukaryotic amyloid proteins, such as Aβ and α-synuclein, indicated by the purple and pink dashed line. The red line indicates proteins such as urease, hemolysins, or PSMα and SCFA can generate reactive astrogliosis, with a pro-inflammatory profile in the cells. The pink dashed line also indicates that amyloid proteins, ureases and catalase can contribute to the increased deposition of α-synuclein.

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
