# Peer review of "Microbial Trojan Horses: Virulence Factors as Key Players in Neurodegenerative Diseases"

_molecules, 2025, doi:10.3390/molecules30030687_

Round 1
Reviewer 1 Report
Comments and Suggestions for Authors
Overall, the review is very comprehensive and detailed, but it is not well organized in terms of section arrangement as different topics are all put under the same Section 2, “Neurodegenerative conditions focused in this review.” Even in this section, I can only see subsections “2.1 Alzheimer's Disease” and “2.3 Parkinson's Disease,” but where/what is subsection 2.2?
I suggest that subsection 2.4 and subsection 2.5 should be merged into one section as Section 3 “Microorganisms involved in neurodegenerative diseases” and then list 3.1 “Gut-Brain Axis” and 3.2 “Oral Cavity Bacteria” as subsections.
It would be better to put subsections 2.6, 2.7, and 2.8 together as one big section, for example, Section 4 “Microbial Virulence Components/Factors and Immune Response”. Subsequently re-arrange subsection 2.9 as Section 5 “Development of Potential Treatments”, which will make the review in a clearer order and easy to follow.
Regarding the treatments, how about using specific antibodies directly and testing other biological molecules such as polyphenols?
Author Response
Overall, the review is very comprehensive and detailed, but it is not well organized in terms of section arrangement as different topics are all put under the same Section 2, “Neurodegenerative conditions focused in this review.” Even in this section, I can only see subsections “2.1 Alzheimer's Disease” and “2.3 Parkinson's Disease,” but where/what is subsection 2.2?
Answer: Sections and topics were re-organized as suggested, and subsection 2.2 was renamed to "Parkinson's Disease." Line 162
I suggest that subsection 2.4 and subsection 2.5 should be merged into one section as Section 3 “Microorganisms involved in neurodegenerative diseases” and then list 3.1 “Gut-Brain Axis” and 3.2 “Oral Cavity Bacteria” as subsections.
Answer: Sections and topics were re-organized as suggested, and subsections 2.4 and 2.5 were incorporated into a new section titled "3 Microorganisms Involved in Neurodegenerative Diseases."
It would be better to put subsections 2.6, 2.7, and 2.8 together as one big section, for example, Section 4 “Microbial Virulence Components/Factors and Immune Response”. Subsequently re-arrange subsection 2.9 as Section 5 “Development of Potential Treatments”, which will make the review in a clearer order and easy to follow.
Answer: Sections and topics were re-organized as suggested to: 4. " Immune Response in Neurodegenerative Disease associated with Microbiota Dysfunction" and 5. "Microbial Virulence Components/Factors." Topic 6 was classified as "Development of Potential Treatments."
Regarding the treatments, how about using specific antibodies directly and testing other biological molecules such as polyphenols?
Answer: An explanation regarding this subject was included in lines 1250-1257 and 1279-1300.
Lines 1250-1257: “Studies report, for example in AD, the existence of many barriers in the creation of vaccines, in this case the heterogeneity of AD, the complexity of the immune response and the need for adequate biomarkers to monitor efficacy. Thus, there are two lines of development: first, passive vaccines that involve the administration of preformed antibodies that bind to beta-amyloid, helping to remove it, like monoclonal antibodies [296]. The second line would be active vaccines that aim to induce a robust immune response against antigens associated with AD, such as beta-amyloid. In this second case, they can be included as vaccines for virulence factors [296].”
Lines 1279-1300: “Antimicrobial peptides are molecules naturally produced by the host and exhibit antibacterial, antiviral, antifungal, and antiparasitic properties. The development of these peptides represents a promising therapeutic alternative to antibiotic resistance, as they are less susceptible to resistance mechanisms. Additionally, they can differentiate between the membrane structures of the host and bacteria. Their interaction with the membrane occurs through electrostatic attractions, allowing them to induce alterations in membrane integrity, ultimately leading to cell death [298].
Polyphenols are organic compounds widely found in cereals, fruits, vegetables, coffee, and other foods. These molecules can be classified into coumarins, tannins, phenolic acids, and flavonoids. Additionally, polyphenols can act as prebiotics, as they are resistant to host digestion and interact with the intestinal microbiota [299,300].
Among the main benefits of polyphenols are their antioxidant and anti-inflammatory properties. They can also inhibit bacterial biofilm formation by interfering with the quorum sensing mechanism, mediated by acyl-homoserine lactone [299,300].
In addition to promoting gut health, polyphenols contribute to microbiota balance by stimulating the growth of probiotic bacteria, such as Bifidobacteriaceae and Lactobacillaceae, while reducing the presence of pathogenic bacteria, such as E. coli, Clostridium perfringens, and H. pylori [299,300].
In the nervous system, polyphenols regulate NF-κB signaling, prevent the formation of toxic Aβ oligomers, and modulate Tau protein hyperphosphorylation, thereby reducing the formation of neurofibrillary tangles and attenuating neurodegenerative processes [299,300].”
We are grateful to both reviewers for the critics and suggestions that improved the quality of our work.
Reviewer 2 Report
Comments and Suggestions for Authors
In this work authors described the fact that microbiota imbalances can contribute to the progression of neurodegenerative disorders by altering the intestinal permeability. The work is well structured and exhaustively described. However, some minor criticisms can be done.
Comments to Authors
In this work authors described the fact that microbiota imbalances can contribute to the progression of neurodegenerative disorders by altering the intestinal permeability. The work is well structured and exhaustively described. However, some minor criticisms can be done as follows.
Line 56. Bacterial extracellular vesicles. Can the authors better explain what are these structures and what is their origin?
Line 75. Authors should better explain what are the changes in glial cells of AD.
Line 382. In this sentence, authors should better clarify the link between the pathogen (H. pylori) and AD.
Line 406. Since Chlamydia pneumoniae mainly colonizes the respiratory tract, how do you explain its involvement in the alteration of the gut-brain axis and the consequent pathogenesis of neurodegenerative diseases?
Line 473. “Inflammation can be directed...”. This concept should be explored further.
Line 532. In the description of the Outer Membrane Vesicles their origins and functions should be better explored.
Line 739. The meaning of pyroptosis should be elucidated.
Line 742. The authors should clarify whether there are any hypotheses on the mechanism of action of LPS and their role in the pathogenesis of neurodegenerative diseases.
Author Response
Comments to Authors
In this work authors described the fact that microbiota imbalances can contribute to the progression of neurodegenerative disorders by altering the intestinal permeability. The work is well structured and exhaustively described. However, some minor criticisms can be done as follows.
Line 56. Bacterial extracellular vesicles. Can the authors better explain what are these structures and what is their origin?
Answer: An explanation regarding this subject was included in lines 565-585 of the revised manuscript, under the subtopic "5.1 Outer Membrane Vesicles (OMVs)."
Line 75. Authors should better explain what are the changes in glial cells of AD.
Answer: This information was present in the original text. However, we added more details (lines 131-145 revised manuscript) to address the reviewer's comment.
The text now reads:
…………
Other factors are also associated with AD, such as hyperactivity of calcium flow around the plaques (which can be normalized by Aβ inhibition). The presence of Aβ plaques and the increase in intracellular calcium can trigger reactive astrogliosis. The elevation of intracellular calcium activates NADPH oxidase and iNOS, leading to the production of ROS in astrocytes, which may culminate in the death of these cells.
Moreover, reactive astrocytes can inhibit phagocytosis by microglia and contribute to glutamatergic dysregulation. The increase in glutamate exacerbates calcium influx into cells, creating a positive feedback loop that further intensifies glutamate release.
Microglia surrounding Aβ aggregates proliferate but are unable to phagocytize them, resulting in a chronic inflammatory state mediated by interleukins such as IL-1β, IL-6, and TNF-α. These cytokines overactivated adjacent glial cells, perpetuating a pro-inflammatory activation state [38–40]. In the same way as astrocytes can induce microglial activation, the reverse effect can also be observed. Reactive microglia are capable of stimulating astrocytes to become immunologically active, promoting the release of inflammatory mediators such as TNF-α, TGF-β, IL-1β, and IL-6 [38].…………..
Line 382. In this sentence, authors should better clarify the link between the pathogen (H. pylori) and AD.
Answer: To clarify the relationship between the pathogen and Alzheimer's disease (AD), we added the following text in line 397-407:
“H. pylori can supposedly reach the brain through the oral-nasal-olfactory route or retrogradely via the vagus nerve. Another infection mechanism involves the use of "Trojan horses," in which activated monocytes (infected by H. pylori due to defective autophagy) can access the CNS through disruptions in the blood-brain barrier.
Metabolites of H. pylori can also compromise the CNS, such as the case of OMVs, which have the ability to cross biological barriers. The bacterium, through its virulence factors like the VacA cytotoxin, exerts chemotactic effects and promotes an increase in the pro-inflammatory response, including the production of TNF-α, IL-8, and ROS. These factors can compromise blood-brain barrier integrity and intensify neuroinflammation. Additionally, the pathogen also induces the expression of genes related to AD, such as ApoE2, ApoE4, APP, and TLR-4.”
Line 406. Since Chlamydia pneumoniae mainly colonizes the respiratory tract, how do you explain its involvement in the alteration of the gut-brain axis and the consequent pathogenesis of neurodegenerative diseases?
Answer: To improve readability, we added a section that makes the inclusion of Chlamydiaceae more consistent. For this purpose, we added topic 3.3 "Pulmonary Microorganisms", line 479.
Line 473. “Inflammation can be directed...”. This concept should be explored further.
Answer: To make the expression clearer, we rewrote the text in lines 459-464 as follows: “Inflammation can spread to the CNS through cell-to-cell communication between circulating macrophages and microglia, making them reactive. Another possibility involves the passage of inflammatory factors from the bloodstream across the blood-brain barrier, resulting in neuroinflammation. Additionally, inflammation can also occur via a retrograde pathway through peripheral nerves (via the trigeminal nerve)”.
Line 532. In the description of the Outer Membrane Vesicles their origins and functions should be better explored.
Awswer: To respond to the reviewer's request, we have added the following text in lines 565-591.
…………
“OMVs can be generated through different budding mechanisms: i) Loss of cross-links (covalent and non-covalent) between the outer membrane and the peptidoglycan layer; ii) Accumulation of misfolded proteins and cellular fragments in the periplasmic space, leading to increased turgor pressure; iii) Imbalance of divalent cations (e.g., calcium), which cross-link negatively charged molecules such as LPS. This imbalance allows LPS molecules to cause electrostatic repulsion, generating membrane curvatures that facilitate OMV formation; iv) Accumulation of phospholipids in the outer leaflet of the outer membrane due to a deficiency in the ABC transport complex VacJ/Yrb; v) Deacylation of the lipid A component of LPS.
In addition to budding, microvesicles can also be formed through explosive lysis. Under cellular stress, often induced by phages, there is an increase in endolysin production, leading to the degradation of bacterial peptidoglycans. This degradation causes the cell to round up and eventually explode. The resulting fragments can reorganize and self-assemble into outer-inner membrane vesicles (OIMVs) or explosive outer membrane vesicles (EOMVs).
Regardless of the release mechanism, microvesicles exhibit immunomodulatory properties in the host due to the virulence factors they carry. These characteristics grant OMVs adjuvant properties, making them potential candidates for vaccine-related applications. Furthermore, OMVs can fuse with bacterial membranes, endowing them with important functional properties such as horizontal gene transfer, cellular metabolite export, phage infection, and cell-to-cell communication.
Their protein/lipid composition is derived from the outer membrane of bacteria and the bacterial periplasmic space/cytoplasm, varying according to the OMVs release process. Their composition may contain LPS, enzymes, cytotoxins, and nucleic acids depending on the specific bacterium considered (Figure 3). In the case of H. pylori’s OMVs, the presence of LPS, urease, the cytotoxins CagA and VacA, and other compounds such as nucleic acids has been reported [132,133]. ………….
Line 739. The meaning of pyroptosis should be elucidated.
Answer: We added a brief explanation about pyroptosis in line 792: “pyroptosis (pro-inflammatory cell death)”.
Line 742. The authors should clarify whether there are any hypotheses on the mechanism of action of LPS and their role in the pathogenesis of neurodegenerative diseases.
Answer: In order to clarify the relationship between LPS and neurodegenerative diseases, we made changes to the text at lines 783-787 and 802-807.
Lines 783-787:
“Once activated, microglia can induce astrocyte reactivity, exacerbating the inflammatory response. This activation triggers multiple pro-inflammatory pathways, including interleukins (IL-1, IL-6, IL-12, IL-17, IL-18, p40, IL-1β, and TNF-α), cytokines (CCL2, CCL5, and CXCL8), and complement system proteins (C3, C3a, and C5a receptor). These factors contribute to increased production of ROS and nitric oxide synthase (NOS) [182]”
Lines 802-807:
“In addition to compromising the blood-brain barrier and promoting a pro-inflammatory response that exacerbates neurodegeneration, LPS can also contribute to the formation of neurofibrillary tangles, enhance Aβ influx into the CNS, impair its clearance, and induce cell death through cyclooxygenase 2 (COX-2) and ERK kinase pathways [182].”